# The neural basis of resting-state fMRI functional connectivity in fronto-limbic circuits revealed by chemogenetic manipulation

Catherine Elorette[1,2,6], Atsushi Fujimoto [1,2,6], Frederic M. Stoll [1,2], Satoka H. Fujimoto [1,2], Niranjana Bienkowska[1,2], Liza London[1,2], Lazar Fleysher[3], Brian E. Russ [1,4,5,7] ✉ & Peter H. Rudebeck [1,2,7] ✉

Measures of fMRI resting-state functional connectivity (rs-FC) are an essential tool for basic and clinical investigations of fronto-limbic circuits. Understanding the relationship between rs-FC and the underlying patterns of neural activity in these circuits is therefore vital. Here we introduced inhibitory designer receptors exclusively activated by designer drugs (DREADDs) into the amygdala of two male macaques. We evaluated the causal effect of activating the DREADD receptors on rs-FC and neural activity within circuits connecting amygdala and frontal cortex. Activating the inhibitory DREADD increased rs-FC between amygdala and ventrolateral prefrontal cortex. Neurophysiological recordings revealed that the DREADD-induced increase in fMRI rs-FC was associated with increased local field potential coherency in the alpha band (6.5–14.5 Hz) between amygdala and ventrolateral prefrontal cortex. Thus, our multi-modal approach reveals the specific signature of neuronal activity that underlies rs-FC in fronto-limbic circuits.

Resting-state functional connectivity (rs-FC) is defined as the temporal correlation of activity signals between brain regions in the absence of a stimulus or task[1]. In the three decades since the invention of functional magnetic resonance imaging (fMRI), and the development of resting-state analyses, rs-FC has become a standard tool for examining network-level functional variations. In particular, it has been used to track variation in frontal circuits in development[2], aging[3], as well as neurological and psychiatric disease[4]. Indeed, in psychiatric research, patterns of rs-FC in fronto-limbic circuits have become increasingly used as biomarkers of subtypes of depression[5,6], autism[7], anxiety[8], and schizophrenia[9]. The validation and use of these biomarkers hold the

potential to enable personalized treatments for subtypes of psychiatric disorders[10,11].

Despite rs-FC's widespread use in basic and clinical research, the neural basis of this measure in fronto-limbic circuits is not clear. In sensory cortex, measures of fMRI rs-FC correlate with the temporal similarity or coherence of neural activity between distributed brain areas[12–16]. Specifically, several studies have observed a close relationship between fMRI rs-FC and local field potential (LFP) coherence in low-frequency bands[13,16–19], leading to the widely accepted notion that this neural mechanism underlies rs-FC across the whole brain, including fronto-limbic circuits. However, fMRI resting-state networks

[1]Nash Family Department of Neuroscience and Friedman Brain Institute, Icahn School of Medicine at Mount Sinai, One Gustave L. Levy Place, New York, NY 10029, USA. [2]Lipschultz Center for Cognitive Neuroscience, Icahn School of Medicine at Mount Sinai, One Gustave L. Levy Place, New York, NY 10029, USA. [3]BioMedical Engineering and Imaging Institute, Icahn School of Medicine at Mount Sinai, One Gustave L. Levy Place, New York, NY 10029, USA. [4]Center for Biomedical Imaging and Neuromodulation, Nathan Kline Institute, 140 Old Orangeburg Road, Orangeburg, NY 10962, USA. [5]Department of Psychiatry, New York University at Langone, 550 1st Avenue, New York, NY 10016, USA. [6]These authors contributed equally: Catherine Elorette, Atsushi Fujimoto. [7]These authors jointly supervised this work: Brian E. Russ, Peter H. Rudebeck. ✉e-mail: brian.russ@nki.rfmh.org; peter.rudebeck@mssm.edu

as well as neurophysiology indicate that frontal areas exhibit functional timescales distinct from those in sensory areas[20,21]. This calls into question whether neural markers of rs-FC observed in sensory areas can be extrapolated to interactions between frontal cortex and limbic areas. In addition, many of the prior investigations of the basis of rs-FC focused on correlating fMRI signals with neural activity markers of functional communication as indexed by LFP coherence, but did not assess how perturbing neural activity altered this relationship[12,15,17,22,23]. Consequently, it is not known if manipulating neural activity in one part of the limbic system is associated with changes in rs-FC and corresponding changes in LFP coherency.

Here we sought to determine the basis of rs-FC in fronto-limbic circuits using a multi-modal approach wherein we causally manipulated neural activity using Designer Receptors Exclusively Activated by Designer Drugs (DREADDs, Fig. 1)[24]. Such chemogenetic approaches enable the reversible control of specific neuronal populations, can be combined with fMRI as well as neurophysiology, and are thus highly amenable for studying the basis of rs-FC. As a relatively new technology, there is conflicting evidence about the overall effect of DREADD-mediated inactivation at the level of brain circuits. A previous macaque study found network-level decreases in rs-FC after amygdala inactivation using clozapine-N-oxide (CNO)[25]. However, more recent rodent research found a paradoxical increase in rs-FC following inactivation of the prefrontal cortex[26]. Using more refined techniques, we aim to reconcile these differences in the macaque. Because of its central role in a host of psychiatric disorders, we specifically targeted inhibitory DREADDs (hM4Di) to the basolateral nucleus of the amygdala. We then assessed how manipulating activity in this area altered rs-FC and LFP coherency in fronto-limbic circuits. To aid selective manipulation of DREADD-expressing cells and assess the causal influence of their

projections across the brain, we used deschloroclozapine (DCZ). This actuator is proven to be inert and has no effect on whole-brain functional connectivity[27,28] or on various types of behaviors[29–31]. We report that DREADD-mediated inhibition of the amygdala increased the brain-wide rs-FC of this area, and specifically increased rs-FC between amygdala and frontal cortex. This increase in amygdala-frontal rs-FC was associated with increased LFP coherence, in specific frequency bands. Our data provide direct evidence for a specific neural mechanism that underlies rs-FC in the circuits that connect the limbic system and frontal cortex.

## Results

### Viral transfection of hM4Di DREADDs produced extensive coverage of amygdala nuclei

To examine the relationship between measures of fMRI rs-FC and neural activity within fronto-limbic circuitry after focal inhibition of amygdala activity, we used a chemogenetic approach (Fig. 1A). Two male macaques underwent neurosurgery to transfect DREADD receptors[24] into amygdala. We injected an AAV vector encoding an inhibitory DREADD (hM4Di) for pan-neuronal expression with a hemagglutinin (HA) marker protein. Virus injections were made bilaterally at 6 sites (3 μl each) targeting the basolateral division of the amygdala[32,33].

Immunohistochemical analysis targeted to the HA marker protein revealed robust expression throughout the amygdala (Fig. 2A). We performed stereological analysis of both animals' amygdala, using sections double-stained for the HA marker protein and Nissl. We observed DREADD labeling of approximately 5% of amygdala cells (Animal H 4.99% cells positive for both HA and Nissl stain compared to just Nissl stain; Animal L 5.21%). In both animals, the strongest labeling occurred in basal and basal accessory nuclei. Both animals additionally

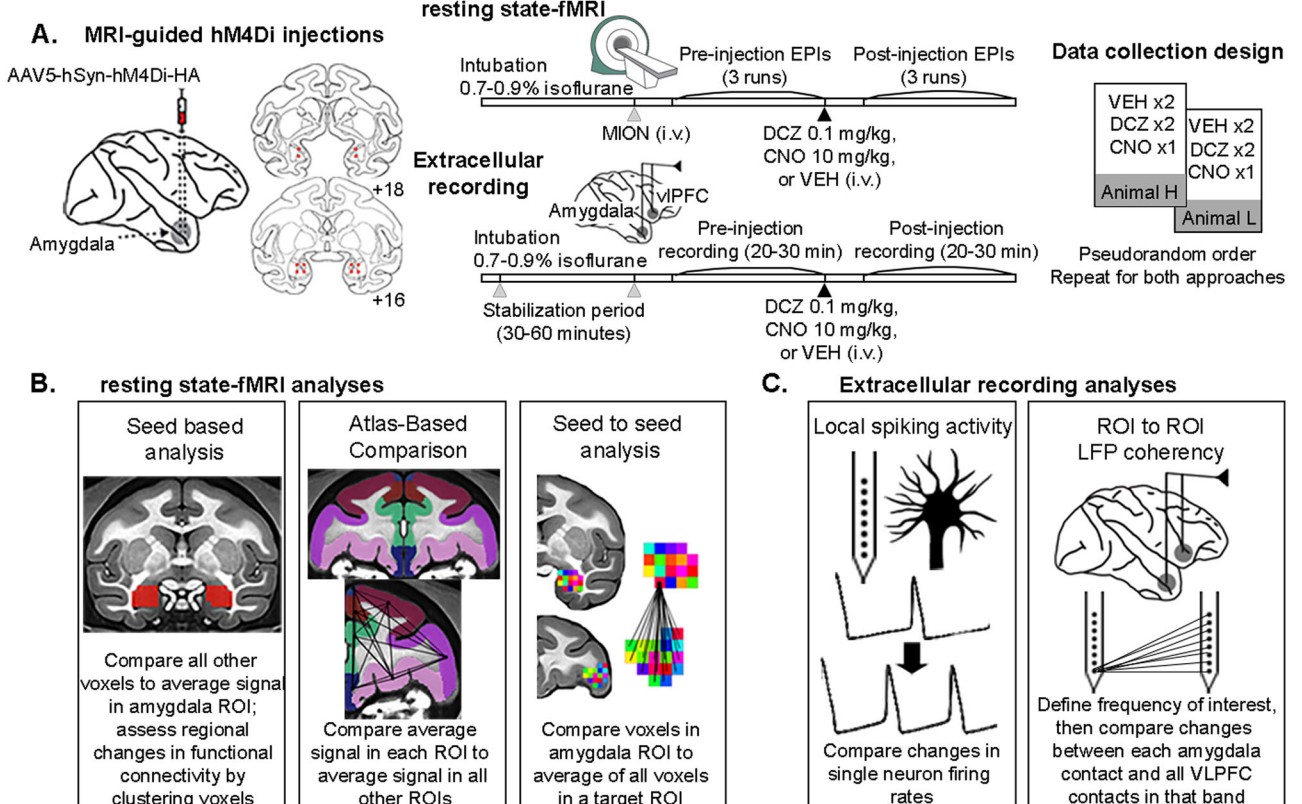

**Fig. 1 | Schematic of experimental design and analysis approach. A** Experimental timeline from injection of DREADDs into amygdala, through data collection of fMRI and extracellular recordings. *VEH = vehicle; DCZ = deschloroclozapine; CNO = clozapine-N-oxide.* Types of (**B**) fMRI or (**C**) neural activity analyses performed. Sagittal and coronal drawings of the macaque brain are taken from a standard rhesus macaque brain atlas (generated by the Laboratory of Neuropsychology, NIMH; in the public domain). Macaque MRI templates shown are taken from the NMT and CHARM atlases[45].

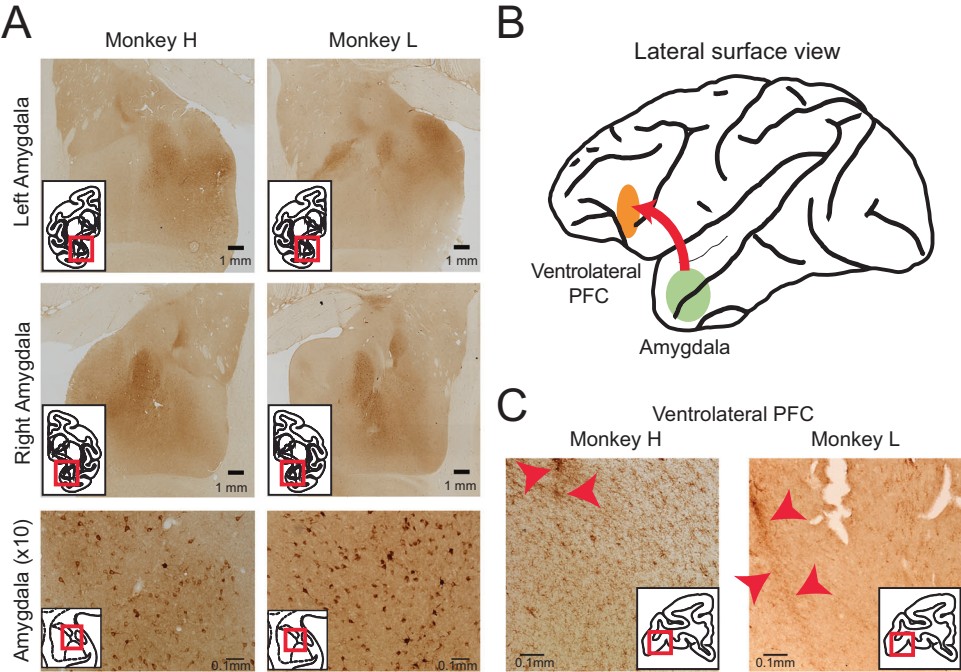

**Fig. 2 | DREADD transfection confirmed by histological processing. A** Brain sections processed for HA tag showing representative images shown of whole amygdala (tiled images, top and middle) as well as DREADD transfected amygdala neurons (bottom). At least 14 sections through the amygdala were stained for each animal. A further 11 sections from each animal were dual-stained for Nissl and DREADDs. **B** Schematic of DREADD injection site in amygdala and labeling of axonal projections in vlPFC. Sagittal drawing of the macaque brain taken from a standard rhesus macaque brain atlas (generated by the Laboratory of Neuropsychology, NIMH; in the public domain). **C** Brain sections processed for HA tag, showing labeling of axon terminals in vlPFC (see inset). Representative images are shown for each animal of the hemisphere in which extracellular recording of vlPFC occurred. Red arrows indicate electrode tracks. At least 6 sections through the vlPFC were stained for each animal.

displayed labeling of cell bodies and axon terminals in the lateral nucleus, which varied by animal and by hemisphere. Specifically, animal H displayed a pattern of labeling that was concentrated medially and anteriorly within the amygdala. The periamygdaloid complex was well labeled in both hemispheres. Some sparse labeling of cell bodies was observed in central nucleus in the right hemisphere. In animal L, labeling was more medial in the left hemisphere and more lateral in the right hemisphere. In this animal, the left periamygdaloid complex showed strong labeling, while the right lateral and central nuclei were also well labeled. When sections throughout the amygdala were co-stained for DREADD receptors and inhibitory or excitatory neuronal markers, we found that inhibitory DREADD receptors were expressed in both types of amygdala neurons (Supplementary Fig. 1). Labeled axon terminals were visible in both animals throughout the amygdala, as well as in regions known to receive projections from basal amygdala, such as the ventral frontal cortex (Fig. 2C). Overall, DREADD receptors were expressed bilaterally in the amygdala in both monkeys. The greatest overlap in expression pattern between subjects was in our target, the basal nucleus, which contains cortical projection neurons[34,35].

### Increased amygdala rs-FC after chemogenetic inhibition

To determine the effects of chemogenetic inhibition of the amygdala on resting state networks in macaques, both animals underwent functional neuroimaging after amygdala DREADDs transfection. Following our previous imaging protocol, we maintained animals on light isoflurane anesthesia (0.7–0.9%)[27]. In each session, functional scans were collected before (pre-injection) and after (post-injection) intravenous administration of either vehicle (VEH) or deschloroclozapine (DCZ) 0.1 mg/kg[29]. Scans collected during the pre-injection period served as within-session baseline data, allowing us to determine the immediate within-session effects of DREADD activation on rs-FC.

We first conducted a region of interest (ROI) analysis to examine the effect of chemogenetic inhibition of amygdala neurons on changes in rs-FC with this area. We used a predetermined anatomy-based ROI for the amygdala that comprised all subnuclei, from the D99 atlas[36] (Fig. 3A). We then separately computed the average correlation between this ROI and every other voxel in the brain collected during the pre-injection and post-injection resting-state scans. Figure 3B shows the rs-FC correlation map from a representative pre-injection scan. In all our pre-injection scans, we found that the amygdala ROIs consistently exhibited strong FC ($p < 0.05$) with the medial prefrontal cortex, ventrolateral prefrontal cortex (vlPFC), and temporal lobes (see Supplementary Fig. 2). This mirrors prior findings[37–39] and known anatomical connectivity[34,40].

To determine changes in rs-FC that specifically occurred as a result of hM4Di-mediated inhibition, we first subtracted the pre-injection amygdala ROI correlation map of rs-FC (see above) from those collected post-injection of vehicle or DCZ. Figure 3C shows an example of the post-injection rs-FC correlation map after treatment with DCZ, demonstrating the effects of activating the inhibitory DREADD receptors on the amygdala rs-FC. Subtracting the pre-injection from post-injection data isolated the change in rs-FC resulting from the injection of vehicle or DCZ, as it reduces the influence of session-to-session scanner variability[27]. Note that our prior work has shown that DCZ alone (at the dose level used here) does not alter rs-FC[27]. We then assessed the effect of DCZ compared to vehicle treatment by subtracting the rs-FC change after vehicle was injected from the rs-FC change after DCZ was injected ([DCZ (post-injection – pre-injection)] – [vehicle (post-injection – pre-injection)]).

The results of the subtraction of rs-FC due to vehicle from DCZ for one animal are shown in Fig. 3D (more examples shown in Supplementary Fig. 3). We observed consistent effects across both animals in regions where rs-FC was increased after DREADD activation with DCZ

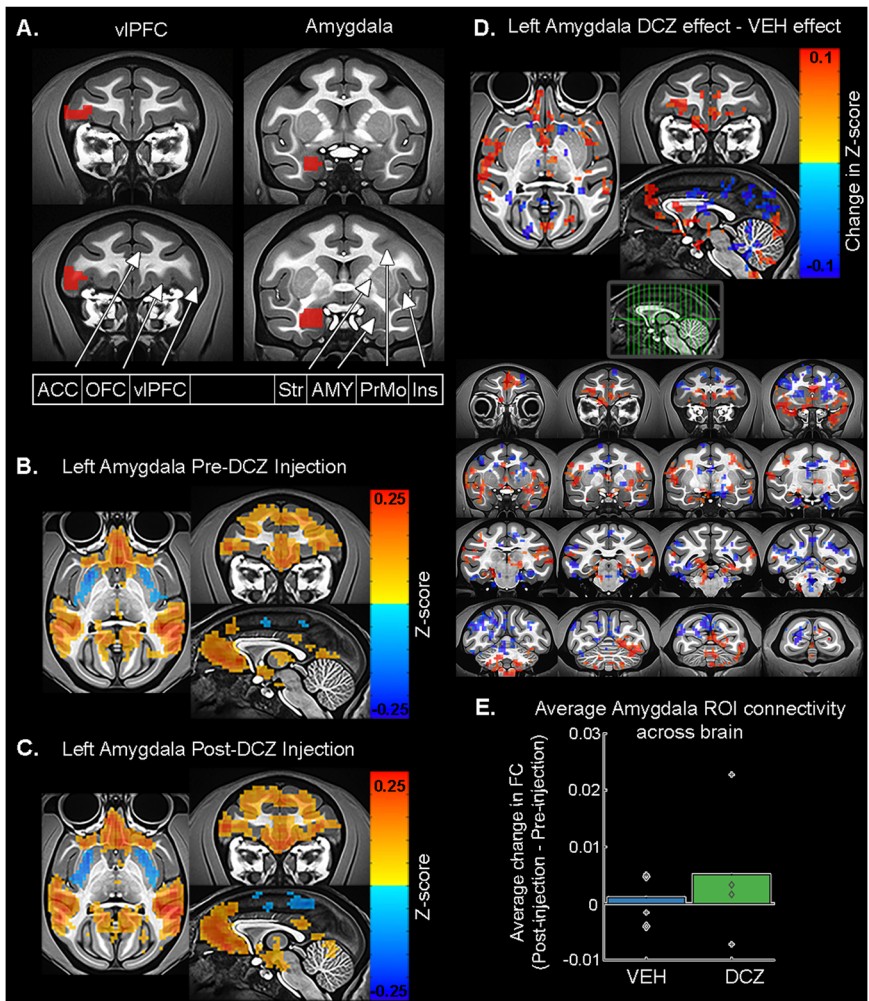

**Fig. 3 | Representative changes in functional connectivity between amygdala and frontal cortex. A** Coronal view of the standardized macaque template[45] showing vlPFC (left) and amygdala (right) ROIs, taken from the D99 atlas[36], high-lighted on the left hemisphere. Select anatomical regions are labeled on the right hemisphere. **B–D** Average rs-FC with a left amygdala seed region in animal L. Pre-DCZ injection period (**B**) and post-injection period (**C**). Scale bar indicates z-score of rs-FC. Threshold p = 0.0485, cluster size ≥30 voxels, voxel faces touching, average of two sessions. r-value range −0.222:0.772 (Pre-DCZ), −0.216:0.794 (Post-DCZ). This analysis was conducted for both hemispheres in both animals (N = 2). **D** Difference in FC produced by DREADD inhibition [(DCZ post-injection − pre-injection) − (vehicle post-injection − pre-injection)], calculated from averaged DCZ post-injection − pre-injection data and averaged vehicle post-injection − pre-injection data. Scale bar indicates difference in z-score of rs-FC. Threshold

p = 0.0485, cluster size ≥30 voxels, voxel faces touching. r-value range −0.205:0.215. This analysis was conducted for both hemispheres in both animals (N = 2). **E** Average (±SEM) change in rs-FC between amygdala (bilateral ROI) and all other voxels in the brain after treatment with vehicle or DREADD activation via DCZ. Symbols indicate individual session average values for each animal (total 226,783 voxel correlations, 2 sessions per treatment, N = 2). Animal H, diamond; animal L, plus sign. Multiway ANOVA, main effect of drug F[1,226780] = 272.20, p = 4.09e-61, main effect of animal F[1,226780] = 186.82, p = 1.64e-42, interaction of drug and animal F[1,226780] = 64.26, p = 1.10e-15. Results are shown throughout on a standard macaque MRI template[45]. Source data are provided as a source data file. *ACC = anterior cingulate cortex; OFC = orbito-frontal cortex; vlPFC = ventrolateral prefrontal cortex; Str = striatum; AMY = amyg-dala; PrMo = premotor cortex; Ins = insula; DCZ = deschloroclozapine; VEH = vehicle.*

compared to vehicle (e.g., anterior cingulate cortex, orbitofrontal cortex, ventrolateral prefrontal cortex, insula, premotor cortex, hip-pocampus, superior temporal gyrus) as well as regions where rs-FC was decreased (globus pallidus, caudate tail, inferior frontal gyrus, middle temporal gyrus, V1, V2). Overall, there was a global change in rs-FC between the amygdala and the rest of the brain (p < 0.0001, Fig. 3E). Our data show that activating the inhibitory-hM4Di DREADD receptors in amygdala caused an increase in rs-FC between amygdala and mul-tiple cortical areas.

The amygdala, including the basal nucleus, projects to both cor-tical and subcortical structures[33,40]. Further, inhibition or lesion of one region can result in widespread changes in network connectivity, even across regions that lack anatomical connectivity to the target[41,42]. Thus, we next sought to dissociate the effect of DREADD-mediated amygdala inhibition on global rs-FC connectomes across: (1) the whole brain, (2)

cortex, and (3) subcortical areas. We assessed rs-FC changes using 3 sets of ROIs extracted from the whole brain atlas (D99), a cortical hierarchical atlas (CHARM, Level 3), and a subcortical hierarchical atlas (SARM, Level 3)[36,43–45]. Importantly, these analyses do not describe the functional connectivity of the amygdala alone, but rather generate a connectome in which every ROI's correlation to every other ROI is measured and the average calculated. This connectome depicts the changes produced in large-scale, brain-wide connectivity as a result of perturbing local activity. It is possible, therefore, that the observed amygdala-frontal cortex rs-FC changes may be influenced by these large-scale network changes.

These connectome analyses revealed that on average activation of the inhibitory DREADD receptors caused a pervasive effect on rs-FC irrespective of whether regions were cortical or subcortical (Fig. 4). Specifically, DCZ was associated with an increase in the rs-FC across

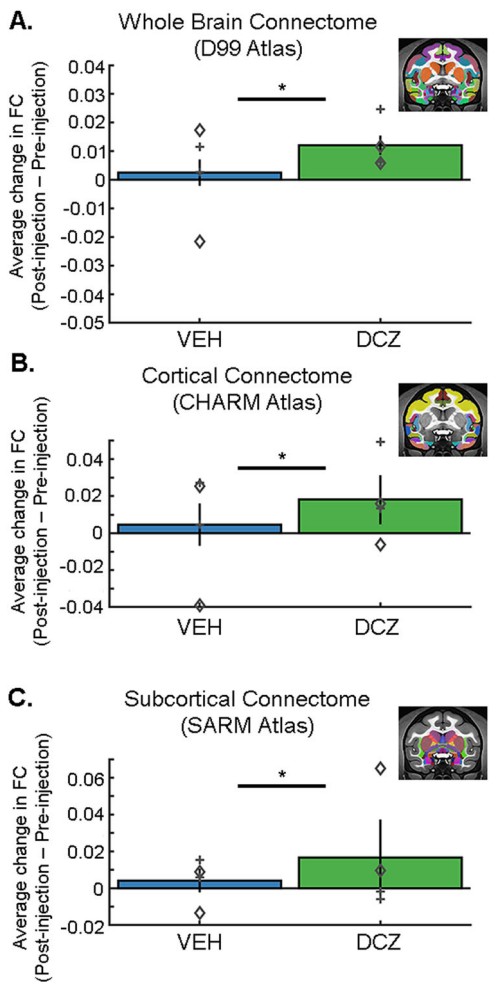

**Fig. 4 | Whole brain fMRI functional connectome altered by chemogenetic inhibition of amygdala.** Average FC across all ROIs calculated across a standard whole brain atlas (**A**), a cortical atlas (**B**) and a subcortical (**C**) atlas. Activation of DREADDs via DCZ increased global rs-FC across all three networks. Symbols denote average difference in rs-FC for each animal, separated by session (2 sessions per treatment, N = 2 animals). Animal H, diamond; animal L, plus sign. Error bars represent SEM. Atlas image insets shown on NMT v2[36,43–45]. Multi-way ANOVA analyses: Whole brain (**A**), main effect of drug $F[1,146,684] = 728.35$, $p = 5.05e\text{-}160$ and subject $F[1,146,684] = 498.91$, $p = 2.51e\text{-}110$, interaction between subject and drug $F[1,146,684] = 12.54$, $p = 0.0004$. Total 146688 voxel correlations. Cortical atlas (**B**), main effect of drug $F[1,5036] = 34.71$, $p = 4.07e\text{-}09$ and subject $F[1,5036] = 115.59$, $p = 1.13e\text{-}26$. Total 5040 voxel correlations. Subcortical atlas (**C**), main effect of drug $F[1,4484] = 44.61$, $p = 2.69e\text{-}11$ and subject $F[1,4484] = 56.76$, $p = 5.92e\text{-}14$, interaction between subject and drug $F[1,4484] = 206.47$, $p = 8.29e\text{-}46$. Total 4488 voxel correlations. Source data are provided as a source data file. *DCZ = des-chlorocolozapine; VEH = vehicle.*

the whole brain (D99, Fig. 4A), a finding that held true when restricted to only cortical (CHARM, Fig. 4B) or subcortical structures (SARM, Fig. 4C). Notably, there were differences between the subjects in each of the three analyses, with a significant interaction between subject and treatment for the whole brain and subcortical rs-FC analyses only. In subcortical structures the two animals had divergent patterns; activating the inhibitory DREADD receptors caused an increase in average rs-FC in animal H, whereas animal L showed a decrease on average (Fig. 4C, symbols). Such a difference may be related to variation in expression of the DREADD receptors in amygdala (Fig. 2; see discussion below). Overall, these analyses reveal that DREADD-mediated inhibition of amygdala neurons is associated with an increase in rs-FC both within specific circuits between the amygdala

and cortical regions, and across networks spanning the whole brain, with the most consistent results seen in cortical networks.

## DREADD-mediated changes in local circuit activity and LFP coherency

There are two possible reasons that the inhibitory DREADD receptors in the amygdala produced an increase in fMRI rs-FC. One possibility is that activating the inhibitory DREADD receptors reduces local neural activity, causing an attenuation of high frequency activity. Consequently, slow oscillations would dominate both local neural activity and the fMRI signals, resulting in more temporally correlated activity between distributed brain areas. Such a pattern would be consistent with work in rodents[26]. Alternatively, activating the inhibitory DREADD receptors may cause an increase in local neural activity in amygdala through recurrent mechanisms, such as the release of excitatory neurons from lateral inhibition and/or inhibition of inhibitory interneurons. The observed increase in rs-FC would therefore be the result of this increased activity driving long-range functional interactions. To determine the relationship between fMRI rs-FC, an indirect measure of correlated neural activity, and direct measures of neural activity, we conducted extracellular neural recordings in our resting-state paradigm in the same animals (Fig. 1C).

Both animals underwent electrophysiology recordings using 2 or 3 16-channel linear arrays simultaneously targeting the amygdala and vlPFC (Fig. 5A). We focused on the relationship between amygdala and the vlPFC (Walker's area 12, located laterally to the lateral orbital sulcus on the ventral surface of the brain) because: (1) the amygdala sends direct mono-synaptic input to vlPFC[34]; (2) this pathway is functionally engaged during reward-based decision making[46,47]; and (3) we found that rs-FC was increased between amygdala and vlPFC after activating the DREADDs (Fig. 3). The recording sites in amygdala and vlPFC were therefore determined based on the prior fMRI analyses (Figs. 2, 3). After recordings, histological analysis confirmed that amygdala neurons expressing DREADD receptors in the two subjects were projecting to vlPFC (Fig. 2C). In order to compare the results across modalities, neurophysiology data were acquired under conditions identical to those used to acquire the fMRI data (i.e., under light isoflurane anesthesia[27]).

First, we analyzed neuronal spiking activity in both recorded areas to establish whether the observed rs-FC changes were directly associated with changes in firing rate. We recorded the activity of 99 single neurons in amygdala across the pre- and post-injection periods for DCZ or vehicle treatment (see Table 1; Supplementary Fig. 4). Activating the inhibitory DREADD receptors with DCZ increased the firing rate of the recorded amygdala neurons compared to the vehicle treatment (Fig. 5B). Additionally, only amygdala neurons after DCZ treatment significantly altered their firing rate compared to the pre-injection period. In the vlPFC, DCZ did not alter firing rate compared to vehicle, and neither treatment altered firing rate from the pre-injection period (Fig. 5C; see methods for details on firing rate modulation analysis). Of the 46 amygdala neurons we recorded in the vehicle condition, 12 (~26%) were significantly modulated with an increase in firing rate. Conversely, 5 (~11%) decreased their firing rate. After DCZ treatment, we observed that of the 53 amygdala neurons, 32 (~60%) increased, and 6 (~11%) decreased their firing rate. In the vlPFC, 7/27 neurons (~26%) in the vehicle condition increased their firing rate, and 3/27 (~11%) decreased it. Following DCZ, 9/18 vlPFC neurons (50%) increased and 3/18 (~17%) decreased their firing rate. These data indicate that activation of the inhibitory DREADD receptors in amygdala was associated with an increase in neural activity locally, but not at distant locations such as vlPFC, confirming that the activity changes were not simply the result of off-target effects of DCZ.

Next, we investigated changes in LFPs. We first looked at how DREADD receptor activation altered the power oscillations in the LFP in both amygdala and vlPFC. Based on the prior analysis of single

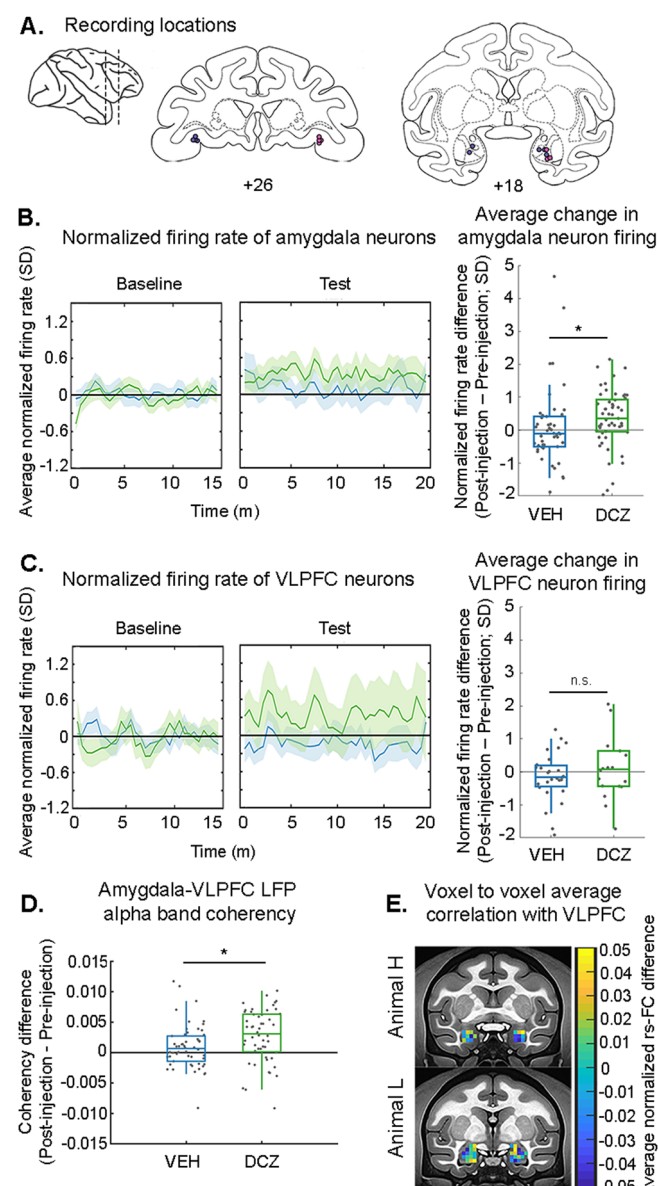

**A. Recording locations**

**B. Normalized firing rate of amygdala neurons**

**C. Normalized firing rate of VLPFC neurons**

**D. Amygdala-VLPFC LFP alpha band coherency**

**E. Voxel to voxel average correlation with VLPFC**

**Fig. 5 | Inhibition of amygdala produces increased functional connectivity as measured by LFP coherency and rs-fMRI, and increased neural spiking.**
**A** Recording sites in vlPFC and amygdala, validated by histology (pink markers, animal H; purple markers, animal L). Sagittal and coronal drawings taken from a standard rhesus macaque brain atlas (Laboratory of Neuropsychology, NIMH; in the public domain). Average time course (left) and summary data (right) of neural spiking activity in the amygdala (**B**) or vlPFC (**C**) after treatment with DREADD actuator DCZ or control. For visualization purposes, the average normalized firing rate change over time was smoothed using a moving average window of 2 s (4 bins). Time course data is displayed as the mean ± SEM. Only amygdala neurons show an increase in average firing rate after chemogenetic inhibition of amygdala via DCZ compared to baseline (Wilcoxon signed-rank test, two-tailed; p = 0.002. Amygdala vehicle p = 0.46; vlFPC DCZ p = 0.68, vlPFC vehicle p = 0.29; amygdala neurons N = 99, vlPFC neurons N = 45). There was a significant difference between amygdala neuron firing rate after treatment with DCZ as compared to vehicle (Kruskal-Wallis; $X^2$ [1,97, N = 99] = 7.3, p = 0.0069), but no difference due to treatment in the vlPFC ($X^2$ [1,43, N = 45] = 0.48, p = 0.49). Single neurons denoted by dots. Box plots display average value (center), 25th–75th percentile data spread (top and bottom of box), and maximal and minimal values (whiskers). **D** Average (±SEM) change in LFP coherency at each amygdala bipolar site with all vlPFC bipolar sites. Dots indicate amygdala bipolar sites (N = 106 across both animals, two recording sessions per treatment). Box plots display average value (center), 25th–75th percentile data spread (top and bottom of box), and maximal and minimal values (whiskers). LFP signal at each bipolar site was band-pass filtered between 0.5 and 200 Hz before a multi-taper frequency transformation was applied; see Methods. Multiway ANOVA, main effect of drug F[1, 102] = 5.64, p = 0.02. **E** Average change in FC between each amygdala ROI voxel with all vlPFC (area 12o/l) voxels, shown separately for each animal, on NMT[45]. Right and left rs-FC maps were calculated separately. Source data are provided as a source data file. *DCZ = deschloroclozapine; VEH = vehicle.*

coherency between amygdala and vlPFC in the alpha band cannot simply be explained by changes in power between drug treatment conditions.

We then extracted the average alpha coherency of each amygdala bipolar site with all vlPFC bipolar sites and for each of these pairs subtracted the pre-injection average coherency value from the post-injection average coherency value. Similar to the normalization procedure conducted on the rs-FC data, this approach reduced the impact of cross-session variability in LFP coherency signals. This analysis revealed a significant increase in alpha-band coherency caused by DCZ-mediated activation of the DREADD receptors compared to vehicle (Fig. 5D). Thus, our results suggest that the DREADD-mediated changes in rs-FC between amygdala and vlPFC were related to increased LFP coherency in the alpha band, in keeping with some prior reports[18].

Finally, because single neuron firing rate changes in vlPFC were variable, we ran a post-hoc analysis to see if the direction of change in firing rate was related to changes in LFP coherency between amygdala and vlPFC on those same bipolar sites. There was no difference in amygdala to vlPFC coherence when vlPFC channels were separated by whether neurons on those channels either increased or decreased their firing rates (Supplementary Fig. 4C, Kruskal-Wallis test, Chi² = 0.47, p = 0.49). This suggests there is not a tight relationship between changes in firing rate and local changes in coherence.

## Comparison of fMRI rs-FC with vlPFC across the spatial extent of the amygdala

The previous analyses indicated that rs-FC between amygdala and vlPFC is related to LFP coherency in the alpha band. We focused our extracellular recordings on the basal nucleus of the amygdala, our original target for DREADD transfection surgery, and on Walker's area 12o of the vlPFC, which receives anatomical projections from the basal nucleus. To compare these results to our fMRI rs-FC findings with more spatial specificity, we conducted a voxel-level analysis of rs-FC between amygdala and vlPFC. We also conducted this analysis to see if there were qualitatively similar effects on rs-FC and the known expression

neuron activity, we might expect that increased neural spiking in amygdala would be associated with increased power in higher frequency bands. Indeed, we observed a statistically significant power increase after DCZ treatment compared to baseline only in the high gamma range (>90 Hz), but this did not differ from the vehicle condition (Supplementary Fig. 5A). We did observe a significant difference in power in the amygdala between DCZ and vehicle treatments between 20 and 60 Hz (within beta and gamma ranges). Power in this range increased from baseline after treatment with vehicle, but either decreased or did not change after DCZ treatment (black dotted line, Supplementary Fig. 5A). No differences in frequencies below 20 Hz between DCZ and vehicle conditions were observed.

Next, we investigated the relationship between LFP activity in our regions of interest by extracting the LFP coherency between amygdala and vlPFC. We first averaged coherograms across bipolar-referenced sites, sessions, and periods, to reveal in an unbiased manner a frequency band of interest for each animal, i.e., a band in which we observed a clear peak in the coherency. For both animals, a peak in coherency was evident from 6.5 to 14.5 Hz, a range centered on the alpha band (Supplementary Fig. 5B). Importantly, the previous power analysis showed there was no change in power within the alpha band between DCZ and vehicle conditions, indicating that any change in

**Table 1 | Summary of recorded neurons**

| | Vehicle | | DCZ | | CNO | |
|---|---|---|---|---|---|---|
| | AMG | vlPFC | AMG | vlPFC | AMG | vlPFC |
| Mk L | 28 | 8 | 31 | 6 | 21 | 7 |
| Mk H | 18 | 19 | 22 | 12 | 7 | 8 |
| Total | 46 | 27 | 53 | 18 | 28 | 15 |

patterns of DREADD receptors in each subject. Thus, these analyses were conducted separately for each animal.

We first calculated the mean connectivity of each amygdala voxel with the vlPFC. Functional connectivity was calculated between each voxel in the vlPFC ROI and each voxel in the amygdala ROI in both the pre-injection and post-injection periods for either DCZ or vehicle (VEH). The pre-injection connectivity matrices were subtracted from the post-injection connectivity matrices, and then the vehicle matrices were subtracted from the DCZ matrices ([DCZ post-injection – pre-injection] – [VEH post-injection – pre-injection]). The correlation values at each amygdala voxel were then averaged such that the resulting values reflect each amygdala voxel's average rs-FC change with all vlPFC voxels following DCZ administration compared to vehicle. We then projected this functional connectivity matrix back onto the amygdala and reasoned that with higher DREADD expression, a subregion of amygdala should exhibit stronger changes in rs-FC with vlPFC (Fig. 5E).

In animal H, dorsal amygdala regions across the medial-lateral extent showed stronger positive rs-FC changes with vlPFC after DREADD activation, while ventral regions showed overall negative rs-FC changes with vlPFC. This pattern of changes appears to align with the DREADD receptor expression patterns in this animal (Fig. 2). In animal L, medial amygdala voxels across the dorsal-ventral extent, as well as voxels in ventrolateral amygdala, showed a positive rs-FC change with vlPFC. Again, the spatial pattern of rs-FC effects appears to mirror the spatial pattern of DREADD receptor expression in this subject's amygdala. In summary, the direction of voxel-based rs-FC between amygdala and vlPFC was relatively heterogenous, suggesting mixed effects at the level of local neuronal circuitry. The strongest positive changes in rs-FC appeared centered on the basal nucleus in both animals, in agreement with our LFP coherency results showing increased coherency between this region and vlPFC.

**Activation of inhibitory DREADDs with clozapine-n-oxide does not reproduce the effects obtained with DCZ**

Throughout this study, we used DCZ, a recently validated compound[29] to activate the DREADD receptors. This actuator is highly selective for DREADD receptors and enables precision targeting with reduced concern for off-target effects on behavior or functional connectivity[27,28,30]. However, much of the initial and current DREADD research uses clozapine-N-oxide (CNO)[24,25]. While CNO has a high affinity for the DREADD receptors, it has been found to have low blood brain barrier penetrance and is back-metabolized to clozapine, causing non-specific effects as clozapine binds to and activates endogenous brain receptors[48,49]. As CNO continues to be used in basic neuroscience, and because CNO was used in an earlier study on the effects of DREADD-mediated inhibition of the amygdala on rs-FC[25], we collected additional fMRI and neurophysiology data where CNO was injected instead of DCZ.

Following CNO, our seed-based fMRI analysis produced an inconsistent pattern of rs-FC between amygdala and frontal cortex such that some regions showed increased rs-FC and others decreased (Fig. 6A). Throughout the brain, treatment with CNO decreased rs-FC with the amygdala ($p < 0.0001$, Fig. 6B). Similarly, the global, atlas-based rs-FC analysis showed, in contrast to DCZ, a significant decrease in overall rs-FC when the DREADD receptors were activated by CNO

(Fig. 6C). This effect was also seen across subcortical, but not cortical, areas.

Using multi-contact linear arrays, we recorded the activity of 28 neurons in amygdala and 15 neurons in vlPFC, as well as LFPs from both areas, before and after intravenous injection of CNO. We observed no significant changes in neural spiking after CNO (Fig. 6D). Despite this, CNO appeared to be associated with more heterogeneous changes in firing rates. CNO significantly modulated a larger proportion of neurons across amygdala and vlPFC areas as compared to vehicle (Fig. 6E). These results suggest that CNO is modulating neural spiking in our target regions. It is, however, producing both increases and decreases in neural spiking to the extent that the overall population activity is not altered. In the frequency band centered on alpha (6.5–14 Hz; described above) we observed an increase in coherence between amygdala and vlPFC following CNO injection (Fig. 6F). This effect was similar to that following DCZ administration (Fig. 5D). Taken together, these results suggest that while both DCZ and CNO activate DREADD receptors, CNO's off target effects cause broad changes in neural circuits and lead to the opposite pattern of rs-FC and neural activity compared to DCZ. These findings further highlight the use of DCZ as an actuator drug for DREADDs and reinforce the importance of controlling for off-target effects when examining changes in rs-FC.

## Discussion

We investigated the neural correlates underlying fMRI rs-FC in fronto-limbic circuits. We used chemogenetics in macaques to transiently alter the activity of amygdala neurons and assessed the effects on fMRI rs-FC and neural activity. We found that activating the inhibitory DREADD receptors was associated with an increase in rs-FC between amygdala and regions that are functionally and anatomically connected to amygdala such as frontal and temporal cortex. We also observed a general increase in functional connectivity across the brain, most notably across cortical regions. When we probed the neurophysiological basis of these effects, we found that chemogenetic inhibition of amygdala neurons produced a modest increase in the amygdala neurons' firing rates. We also characterized the changes in neurophysiological markers of rs-FC between the amygdala and vlPFC, which receives direct amygdala projections. LFP coherence in the alpha band between these brain areas increased after chemogenetic inhibition of the amygdala, similar to the increase in fMRI rs-FC. This pattern of effects provides convergent evidence that coherency in the low frequency bands of the LFP may underlie rs-FC measures in frontal-limbic circuits.

Prior work comparing hemodynamic signals to direct recordings of neuronal activity reported that fMRI rs-FC is closely related to patterns of synchronous or coherent activity in specific LFP frequency bands[50]. However, the specific frequency band that underlies rs-FC has been harder to pinpoint. For example, using electrocorticography in humans, rs-FC between lateral and medial parietal cortex is related to correlations in the low beta and high-gamma bands[51]. By contrast, LFP coherence in the alpha band across macaque pulvinar, lateral intra-parietal area, V4, and TEO correlated most strongly with fMRI rs-FC[18]. Other studies have similarly highlighted the importance of other LFP frequency bands[52,53]. Our experiments revealed that amygdala to frontal cortex LFP coherency was strongest centered on the alpha band (between 6.5 and 14.5 Hz) and agrees with the findings of Wang and colleagues on cortico-cortical rs-FC in macaques[18]. This should not, however, be taken as evidence that coherence in the alpha band is the only mechanism underlying all rs-FC. Other groups have shown that the frequencies that contribute to rs-FC vary depending on the specific circuit or functional network[52,54,55]. Our data reveal that in amygdala-frontal pathways, rs-FC is associated with alpha band coherence, knowledge that is likely important for developing targeted therapies that aim to alter functional networks, a point we take up below.

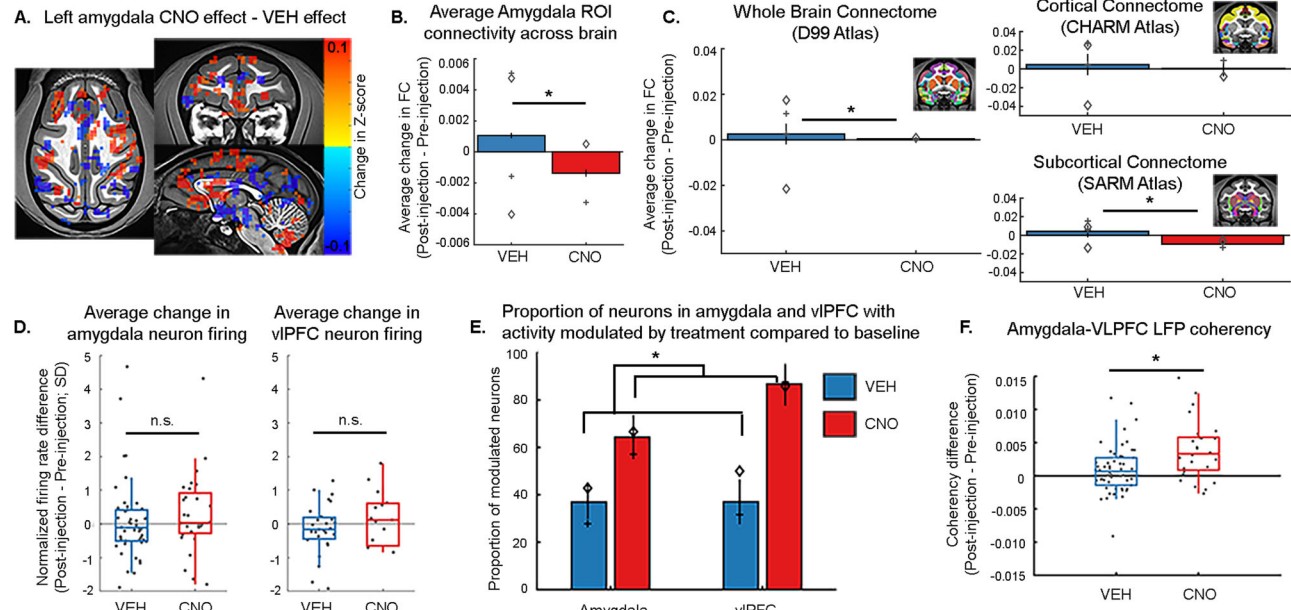

**Fig. 6 | Inhibitory DREADDs activation by less specific actuator CNO produces similar changes in frontal-limbic functional connectivity but differently affects large scale connectomes.** All analyses conducted in two animals (N = 2). **A** Animal L left amygdala ROI FC difference after CNO treatment compared to vehicle, on NMT[45] (as in Fig. 3D). Threshold p = 0.0485, cluster size ≥30 voxels. r-value range −0.304:0.261. Analysis was conducted for both hemispheres. **B** Average (±SEM) change in rs-FC between amygdala (bilateral ROI) and all other voxels after vehicle or CNO treatment. Symbols indicate animal session values: Animal H, diamond; animal L, plus sign. Multiway ANOVA, main effects of drug F[1,170084] = 67.70, p = 1.91e-16, animal F[1,170084] = 16.23, p = 5.61e-5, and interaction F[1,170084] = 76.22, p = 2.56e-18. **C** Average (±SEM) rs-FC calculated using whole brain, cortical, or subcortical atlases. CNO decreased global and subcortical rs-FC. Whole brain multiway ANOVA, main effects of drug F[1,110012] = 24.49, p = 7.48e-07, subject F[1,110012] = 90.89, p = 1.55e-21, and interaction F[1,110012] = 140.66, p = 2.00e-32. Total 110016 voxel correlations. Cortical atlas multiway ANOVA, main effect of subject F[1,3776] = 51.04, p = 1.08e-12, no drug (F[1,3776] = 2.17, p = 0.14), no interaction (F[1,3776] = 1.05, p = 0.31). Total 3780 ROI correlations. Subcortical

atlas, main effect of drug (F[1,3362] = 42.99, p = 6.35e-11), subject and drug interaction (F[1,3362] = 23.28, p = 1.46e-06), no main effect of subject (F[1,3362] = 1.77, p = 0.18). Total 3366 ROI correlations. **D** CNO didn't alter amygdala (left; Kruskal-Wallis; X² [1,72,N = 74] = 2.0, p = 0.16) or vlPFC (right; Kruskal-Wallis; X² [1,40,N = 42] = 0.52, p = 0.47) neural spiking. Dots indicate single neurons. Box plots display average value (center), 25th–75th percentile data spread (top and bottom of box), and maximal and minimal values (whiskers). Across conditions, firing rate was not significantly altered compared to pre-injection (Wilcoxon signed-rank test; amygdala: CNO p = 0.33, vehicle p = 0.46; vlPFC: CNO p = 0.85, vehicle p = 0.29). **E** CNO treatment did significantly increase the proportion of modulated neurons (average ±SEM) across the amygdala (N = 74) and vlPFC (N = 42). Linear mixed-effects model T[1,112] = 2.25, p = 0.03. **F** CNO significantly increased amygdala-vlPFC LFP coherence. Dots indicate coherence with all vlPFC sites at each amygdala bipolar site (N = 82 across both animals, two vehicle, one CNO recording session each). Box plots are the same as Fig. 6D. Multiway ANOVA, main effect of treatment (F[1,78] = 10.73, p = 0.002). Source data are provided in source data file. *CNO = clozapine-N-oxide; VEH = vehicle.*

One important caveat is that the macroscale functional neuroimaging and micro/meso-scale electrophysiological recordings were collected in separate sessions. Simultaneous fMRI and electrophysiological recordings could allow a direct comparison between the modalities, but there are many technical barriers, particularly when considering single-neuron spiking data[56]. However, the use of the same paradigm performed in the same subjects has been previously used to provide a link between fMRI and underlying neural activity[57,58]. We use a within subject design across experiments, allowing for comparison of results across both modalities while reducing possible systematic effects from variability across subjects. Consequently, while it is not possible to directly link specific features of fMRI and neural activity signals (i.e., spiking in amygdala to fMRI fluctuations), our design allows for correlational links between local and systems level effects. Additionally, our work provides a foundation from which future studies may more directly examine the relationship between fMRI rs-FC and neural activity in limbic and frontal areas.

As noted above, most prior work on the basis of rs-FC looked at the correlation between fMRI and neural activity without perturbing activity within a circuit. Doing so would provide evidence for the neural mechanism of rs-FC. Lesions in macaques have revealed that sectioning the corpus callosum was associated with a decrease in intra-hemispheric rs-FC[59]. This work and others using lesions[60,61] were, however, unable to reveal the associated changes in neural activity that

followed the lesion-induced changes in rs-FC. As lesions involve local cell death, these studies cannot assess how increases in local activity might change rs-FC. Aspiration lesions can have off-target effects[62,63], further limiting the insights that can be gained from this approach. To overcome these issues, we used DREADDs to specifically target amygdala neurons. Previous work supports the use of this approach in macaques for studying rs-FC changes[25,42,64]. By successfully combining chemogenetics with fMRI and neurophysiology, our results provide causal evidence that increasing rs-FC between amygdala and vlPFC is directly related to increased synaptic modulation, specifically in the alpha band of the LFP.

Given the similarity between our findings and Wang and colleagues[18] it is likely that alpha band coherence underlies rs-FC in other fronto-limbic or cortico-cortical pathways. Despite this, the effects of chemogenetic modulation of rs-FC and neural activity still are likely to be circuit specific. This difference between neural circuits was highlighted by a recent study in rodents that also used DREADDs to manipulate rs-FC. The authors found a similar relationship between LFP coherence and rs-FC[26]. Specifically, chemogenetically inactivating prelimbic cortex led to an increase in rs-FC and LFP coherence between it and thalamus in the delta band (1–4 Hz). Thus, both ours and the study by Rocchi and colleagues[26] consistently show that rs-FC is directly and causally related to LFP coherence. On the other hand, there are clear differences between the effect of inhibitory DREADDs

on neural activity and the specific LFP frequency underlying rs-FC between the two studies (delta vs alpha). Such variation in findings is likely related to the circuits[55], the species, (mice versus macaques), or some combination of both.

We observed that activating inhibitory DREADDs in the amygdala caused an increase in neural spiking locally, but not in vlPFC. Importantly, chemogenetic studies in macaques, including our own, have not been able to determine if the recorded neurons are those that have expressed DREADD receptors, or are simply communicating with DREADD-expressing neurons. Therefore, our amygdala neurophysiological sampling does not necessarily represent firing changes solely in DREADD-expressing neurons. Further, when recording from a site downstream of the transfected region, such as vlPFC, it is difficult to know how the extent and strength of the projections will affect excitability. We did not observe a relationship between changes in vlPFC spiking and amygdala-vlPFC LFP coherence. We therefore consider our findings as modulations in overall local excitability, compared to direct measures of individually affected neurons. Only three recent macaque studies have recorded spiking activity after DREADD transfection[65–67], and all observed a decrease in firing rate after chemogenetic inhibition. However, these studies used constructs that targeted various cell populations (respectively, all neurons, all cells, and all catecholaminergic neurons), none recorded from amygdala, and none targeted fronto-limbic circuitry. We targeted all neurons, so our transfected cells included inhibitory and excitatory amygdala neurons (Supplementary Fig. 1). This makes our inhibitory chemogenetic approach more comparable to pharmacological inhibition or excitotoxic lesion, but alters the interpretation of the effects. Our finding that activating an inhibitory DREADD in macaque amygdala led to increased local spiking (Fig. 5) could be related to the composition of this structure in primates. There is an increased diversity of interneurons in macaque amygdala compared to rodents[68–70]. Thus, we speculate that inhibiting these cells may cause a paradoxical disinhibition of local circuits, leading to increased extracellular activity. A second possibility is that the inhibition of local excitatory neurons that drive inhibitory networks within the amygdala leads to increased local activity[71]. These explanations are not mutually exclusive, and future work should clarify the mechanisms at play. The use of DREADD constructs specific for excitatory neurons, or targeting specific projections, would be a valuable next step. In summary, our findings and those of Rocchi and colleagues[26] emphasize the importance of confirming the effect of chemogenetic modulation on neural activity, and highlight the power of chemogenetic manipulation for revealing the basis of rs-FC.

To date, few studies have investigated the neural basis of functional connectivity in limbic circuits, either generally or in specific relation to the amygdala. A previous paper examining changes in fMRI rs-FC after DREADD-mediated inhibition of the amygdala found decreased FC between amygdala and frontal regions after administration of the ligand CNO[25]. This result appears to be counter to our findings, namely that DREADD-mediated inhibition of amygdala neurons increased rs-FC. Since Grayson et al.[25] was published, refinements have been made in macaque anesthetized fMRI protocols[27,72,73], DREADD constructs[74], and DREADD actuators[29,48,49].

In part because of these refinements, we collected our data while animals were maintained under light (≤0.9%) isoflurane. Although anesthesia can reduce functional connectivity across resting state networks as dosage increases[75–78], low-level anesthesia is a compromise that prevents motion artifacts while allowing for maximal preservation of resting state network activity[73,79–81]; this is particularly important given that cortical rs-FC is most strongly impacted by increasing doses of anesthesia[75,79,82]. In light of work finding that DREADD constructs carrying an mCherry fusion protein are only sparsely expressed at the cell surface[74], we chose a DREADD construct that contains an HA tag instead. Finally, we used a highly selective, potent DREADD actuator

developed for use in macaques, DCZ, rather than CNO[29]. CNO can be metabolized to clozapine, which has broad affinity for endogenous neuromodulatory receptors in addition to DREADDs[48,49]. Additionally, CNO can affect brain activity during sleep, which may have implications for activity under anesthesia[83].

Although the anesthesia dose and viral vector used for DREADD transfection likely impacted our results, we also reasoned that the off-target effects caused by CNO/clozapine might be the key difference between Grayson's[25] and our studies. Thus, we conducted one session in each animal and experimental paradigm where we administered CNO, rather than DCZ. Administration of CNO, as compared to DCZ, caused a pattern of effects that was more similar to what Grayson and colleagues[25] reported (Fig. 6). Notably, administration of either actuator caused the same increase in LFP coherence between amygdala and vlPFC. Thus, our data show that activating inhibitory DREADD receptors, regardless of actuator, has a consistent effect on neurophysiological markers of rs-FC, but simultaneously that the actuator chosen has different effects on fMRI rs-FC[27,29].

Why CNO activation of DREADD receptors causes a different pattern of rs-FC is an open question, but likely relates to this ligand being metabolized to clozapine[48,49]. Clozapine has a high affinity for D1 and D2 dopamine and 5-HT2A and 5-HT2C serotonin receptors[84] that are widely distributed across the brain. The off-target effects of clozapine on these receptors may underlie the wide-scale effects on rs-FC. Therefore, it may be difficult to interpret the results of functional studies using CNO or clozapine exclusively, as distinguishing the effects of DREADDs from off-target activity is challenging. Further, individual animals likely metabolize CNO to clozapine at differing rates, complicating interpretation. This means that some previously reported behavioral findings using DREADDs receptors may be driven by CNO/clozapine off-target effects. Our results lend further support that studies using DREADDs should use DCZ as an actuator.

In recent years, patterns of rs-FC in fronto-limbic circuits are being explored as potential biomarkers of psychiatric disorders[85]. With the advent of large cohort imaging studies, it has been possible to identify subtypes of depression[5,6], autism[7], anxiety[8], and schizophrenia[9] based on their differential rs-FC patterns. These biomarkers could identify and then target neuromodulation therapies such as deep brain stimulation (DBS) to specific circuits to ameliorate symptoms[86]. Thus, our finding that rs-FC in a fronto-limbic circuit is directly related to coherence in the LFP alpha band is translationally relevant as it establishes the baseline for rs-FC patterns in healthy circuits that are often implicated in psychiatric disorders[87]. Consequently, this information could help optimize treatment approaches such as DBS to attempt to heighten alpha band communication between affected areas.

## Methods

### Subjects

Two male adult rhesus macaques (*Macaca mulatta*), age 7, were used as subjects in this study. Both animals (animal H, animal L) underwent anesthetized functional MRI scans before and after DREADD transfection surgery. Both animals also underwent anesthetized electrophysiological recording from the amygdala and frontal cortex after DREADD transfection surgery. All procedures were reviewed and approved by the Icahn School of Medicine Animal Care and Use Committee.

### Surgical targeting MRI

A pre-operative MRI scan was collected in each animal to target the amygdala. The animal was sedated with a mixture of ketamine (5 mg/kg) and dexmedetomidine (0.0125 mg/kg), intubated, and placed into a standard MRI-compatible stereotaxic frame. Animals were then transported to the imaging facility where they were maintained at a stable plane of anesthesia using isoflurane (1%–4%) during image

acquisition. Between 2 and 4 3D T1-weighted images (0.5 mm isotropic, TR/TE 2500/2.81 ms, flip angle 8°) were acquired on a Siemens MAGNETOM 3 Tesla Skyra scanner using a custom-built 4-channel phased array surface coil with local transmit function (Windmiller-Kolster Scientific, CA).

## Surgical procedures

All surgical procedures were conducted under isoflurane (2–3%) general anesthesia.

First, all animals underwent an AAV vector injection surgery. AAV vector encoding for inhibitory DREADD (AAV5-SYN1-hM4Di-HA 1.7 × 10^13 GC/ml (Addgene, Watertown MA)) was injected into the bilateral basolateral amygdala stereotaxically. pOTTC1484 - pAAV SYN1 HA-hM4D(Gi) was a gift from Christopher Richie (Addgene plasmid #121538; http://n2t.net/addgene:121538; RRID:Addgene_121538). Six sites within the amygdala were chosen to achieve full coverage of the structure. One site was chosen to target basal amygdala, and a further two were located 2 mm posteriorly and targeted basal accessory, basal, and medial nuclei. Each of these sites had an additional site targeted 2 mm ventrally. Injection sites were chosen by consulting each animal's pre-surgical MRI for precise targeting.

To inject the viral vector, first a small craniotomy was made in the skull over each amygdala. The virus was drawn up into a syringe and the surface of the needle was wiped down with alcohol and sterile saline. The syringe was positioned using an arm attachment on the stereotaxic frame, and the needle was lowered into the brain. To reduce both pressure at the injection site and the chance of tissue blockage at the tip of the needle, at each of the three locations the needle was first lowered 2 mm ventral to the target, then pulled up to the correct coordinates. The more ventral site was injected first, then the more dorsal site. Injections were made by hand at an approximate rate of 0.2 μl/min. After each injection, at least five minutes elapsed before the needle was moved to reduce any leakage of virus. 3 μl were injected at each site for a total of 18 μl per hemisphere. At least two months elapsed after surgery before any experiments were conducted to allow for full viral expression.

Once neuroimaging experiments were completed, a rectangular acrylic chamber[88] was implanted with ceramic screws (Thomas Recording, Giessen, Germany) and biocompatible acrylic (Lang Dental, Wheeling, IL), which allowed access to both the amygdala and ventrolateral prefrontal cortex. Following a 2-week recovery period, a craniotomy was made inside the chamber. Coordinates for surgical planning were determined with a pre-acquired T1-weighted MR image.

## Drug preparation

Deschloroclozapine (DCZ) solution was prepared using DCZ powder (HY-42110, MedChemExpress, Monmouth Junction, NJ), dissolved in dimethyl sulfoxide (DMSO, 2% of total volume) and then diluted in saline to a concentration of 0.1 mg/kg[29], total volume of 1 ml. Clozapine-N-oxide (CNO) solution was prepared using CNO powder (Jin lab, Mount Sinai School of Medicine), dissolved in dimethyl sulfoxide (DMSO, 2% of total volume), and then diluted in saline to a concentration of 10 mg/kg, total volume 10 ml. Vehicle solution was prepared as 2% DMSO diluted in 1 ml saline. All solutions were prepared within 30 min of usage.

## fMRI data acquisition

We used a protocol for functional imaging that our group has previously successfully applied[27]. Animals were first sedated with ketamine (5 mg/kg) and dexmedetomidine (0.0125 mg/kg) 1–1.5 h before the data collection to prevent detrimental effects of ketamine on neural activity. Animals were placed inside an MRI-safe plexiglass primate chair in the sphinx position during image acquisition (Rogue Research, Cambridge, MA). The animal's head was not restrained, but supported with towels and flexible bandages to allow for stability with

minimal discomfort. This setup allowed for the use of lower maintenance isoflurane levels during image acquisition; isoflurane anesthesia was maintained at 0.7–0.9% throughout all functional imaging sessions. The use of low-level anesthesia allows for preservation of resting-state networks[81,82,89]. To minimize the effect of physiological changes in the neural activity, vital signs (end-tidal CO2, body temperature, blood pressure, capnograph) were continuously monitored and maintained throughout an experimental session.

In each session, pre-injection and post-injection (vehicle, DCZ, CNO) functional scans were collected. First, a set of setup scans were acquired, which included shimming based on the acquired fieldmap. A session specific 3D T1-weighted image (0.5 mm isotropic, TR/TE 2500/2.81 ms, flip angle 8°) was acquired. Following monocrystalline iron oxide nanoparticle (MION) injection (i.v.)[90,91], three functional scans (Echo Planar Images (EPI): 1.6 mm isotropic, TR/TE 2120/16 ms, flip angle 45°, 300 volumes per each run) were obtained for pre-injection functional resting states. Prior to DREADD surgery, two sessions were performed with each animal in which this was the only data collected (see Supplementary Fig. 6). Next, either vehicle, DCZ (0.1 mg/kg), or CNO (10 mg/kg) was administered (i.v.). In sessions where DCZ or vehicle were tested, we allowed 15 min after the injection to ensure trafficking of the drug to the brain[29], prior to the collection of an additional three functional runs. The order of DCZ and vehicle testing was counterbalanced across animals. Each animal completed five imaging sessions in total. Both monkeys H and L completed two sessions each of imaging with vehicle and DCZ injection. Drug sessions were alternated and the order of DCZ and vehicle testing was counterbalanced across animals. After the conclusion of the vehicle and DCZ sessions, both animals completed one session where CNO was administered. In this session, the post-injection EPI data were collected ~30 min after CNO injection to allow for the longer time course of CNO trafficking and metabolic processing[29]. All of the anesthetized neuroimaging conducted here was done using the contrast agent MION, as a number of prior studies in macaques has shown that it improves the quality of rs-FC data in both awake and anesthetized states[25,72,90,92].

The power in this study design is derived from the extended within-session acquisition as well as from the replications within-subject. Previous work has demonstrated that greater than 30 min of resting-state data in NHPs provides a high degree of reliability within an individual subject[73,77,93], however, some variation occurs across sessions. The collection of pre- and post-injection data for each treatment reduces the influence of across-session variability. Collecting replications improves the signal to noise ratio within subjects[80,93]. To reduce multiple comparisons and improve power outside of data acquisition, fMRI results were analyzed using a priori defined ROIs and results were compared to well characterized resting state networks.

## fMRI data analysis

Functional imaging data were preprocessed with standard AFNI/SUMA pipelines (version 21.0.08)[27,45,94]. Raw images were first converted into NIFTI data file format and ordered into BIDS format[95]. The T1-weighted images were spatially normalized, then skull-stripped using the U-Net model built from Primate Data-Exchange open datasets[96]. The normalized skull-stripped T1-weighted image was then aligned to the NMT version 2.0 atlas[43], along with subject-specific versions of the D99[36], Cortical Hierarchical Atlas of the Rhesus Macaque (CHARM)[45] and Subcortical Atlas of the Rhesus Macaque (SARM) atlases[44].

The functional data were preprocessed using a customized version of the AFNI NHP preprocessing pipeline[27,45]. For each session, pre- and post-injection data were processed separately using the same parameters. In addition to a set of two dummy scans, the first two TRs of each EPI were removed to ensure that any magnetization effects were removed from the data before functional connectivity analyses. The images were first slice time corrected, then motion correction was applied, the EPIs were aligned to the within session T1-weighted image,

and warped to the standard space. Following alignment to the standard space, the EPIs were blurred with an FWHM of 3 mm, and then converted to percent signal change. Finally, the motion derivatives from each scan along with cerebrospinal fluid and white matter signal were regressed out of the data. The residuals of this analysis were then used to compute the functional connectivity analysis described below.

To examine the effects of drug on amygdala connectivity specifically, we created an amygdala mask from the standard D99 atlas[36]. We extracted the average time series from this amygdala ROI for each condition in each monkey, before and after drug treatment. We then calculated the Fisher's z-transformed Pearson's correlation between the average amygdala time series and the time series of every voxel in the brain (AFNI's 3dTcorr1D). To statistically evaluate the effect of DCZ on amygdala functional connectivity, correlation maps were submitted to an ANOVA with factors of drug (vehicle or DCZ), and monkey, with session nested under monkey. We also examined connectivity in a more qualitative way by examining correlation maps between the left or right amygdala ROIs and every voxel in the brain. We subtracted the vehicle correlation maps from the DCZ or CNO correlation maps to reveal the change in correlation caused by the drug treatment, and applied a voxelwise FDR correction of $0 = 0.05$, then further corrected these results by applying cluster correction with AFNI's 3DClusterize command. We used a two-tailed approach in which positive and negative r-value voxels were clustered separately, with a minimum amount of 30 voxels that shared at least one side in contact with other voxels in the cluster.

Whole brain connectome analyses were computed using 3dNet-Corr function in AFNI[94,97]. For each animal and each session, we computed the correlation between ROIs of the D99 atlas[36], level three of the CHARM atlas[45], and level three of the SARM atlas[44]. The individual correlation matrices, or connectomes, were then Fisher's z-transformed. For each session, we subtracted the pre-injection connectome from the post-injection connectome, creating a matrix of the functional connectivity changes. To statistically evaluate the effect of vehicle or DCZ on the functional connectivity across ROIs the difference connectomes were submitted to an ANOVA (function anovan in MATLAB) with factors of drug and subject. ROI was modeled as a random effect, and session was nested under subject. We calculated the effect of CNO compared to vehicle separately, as we only obtained one session with CNO. Similar to the above analysis, the difference connectomes were submitted to an ANOVA with factors of drug and subject, with ROI pair modeled as a random effect.

We further examined the relationship between the amygdala and the ventrolateral prefrontal cortex on an individual voxel basis[98]. Using MATLAB (version 2022b) and SPM12, we extracted the individual time series from each voxel in the amygdala ROI and the vlPFC (ROI taken from the CHARM 6 atlas; combined masks for area 12o and area 12l)[45]. Analyses were performed separately for left and right hemispheres. A Fisher's z-transformed correlation map was calculated between each voxel in the ipsilateral amygdala and the vlPFC. The average vlPFC z-value was then calculated for each amygdala voxel. The pre-injection correlation was subtracted from the post-injection matrix, producing a voxel-wise map of the change in functional connectivity between the two regions produced by drug administration.

## Neurophysiology data acquisition

We used the same anesthesia protocol as in the fMRI sessions, to maximize cross-modal comparisons. Following intubation, the head was fixed in a plastic stereotaxic frame and the sedation was maintained with low-level isoflurane throughout the session. In each session, the dura matter was penetrated using a tungsten guide tube (Crist Instruments, Hagerstown, MD) and then two (animal H) or three (animal L) 16-channel linear arrays (S-probe, Plexon, Dallas, TX) were lowered using a NAN microdrive system (NAN instruments, Nazareth, Israel). In both animals, individual electrodes targeted both the

basolateral amygdala and vlPFC in the same hemisphere (monkey H: left hemisphere, monkey L: right hemisphere), and an additional electrode (monkey L) targeted the contralateral amygdala. The recording depth and grid coordinates were calculated based on a high resolution T1w imaging with the animal in an MRI compatible stereotaxic frame. In animal H, the amygdala recording site was 22 mm anterior to the interaural plane, 10 mm to the right of the midline, and the vlPFC recording sites were $29.5 \pm 0.5$ mm anterior to the interaural plane, $14.5 \pm 0.5$ mm to the right of the midline. In animal L, the amygdala recording sites were 19 mm anterior to the interaural plane, $7.5 \pm 0.5$ mm to the left of the midline and 19 mm anterior to the interaural plane, $8.5 \pm 0.5$ mm to the right of the midline, and the vlPFC recording site was 27 mm anterior to the interaural plane, $13.5 \pm 0.5$ mm to the left of the midline.

The electrodes were advanced to the target depths while listening to gray and white matter transitions to identify targets. We allowed neural signals to stabilize at the target depth for 1 h before beginning data collection. Recording sessions consisted of three time periods: pre-injection (30 min), injection (15–30 min), and post-injection periods (30 min). Following the pre-injection period, a drug (DCZ, CNO, or vehicle) was injected intravenously, followed by a waiting period (injection) to allow for the known pharmacodynamics of the drugs in macaque monkeys[29], and was consistent with the waiting period used during fMRI acquisition. The anesthetized recording was performed at least 1 week apart, and the order of drug conditions was counterbalanced across animals.

Wide-band signal was recorded for each electrode contact using an Omniplex system and Plexcontrol software (version 20, Plexon) and stored for offline analysis. Signal was first band-passed (600 to 6000 Hz) before extracting all spike waveforms exceeding 3 sd. Spikes from putative single neurons were then automatically clustered offline using the MountainSort plugin for MountainLab[99] and later curated manually based on principal component analysis, interspike interval distributions, visually differentiated waveforms, and objective cluster measures (Isolation probability >0.75, Noise overlap probability <0.2, Peak signal to noise ratio >0.5 sd, Firing Rate >0.05 Hz). Our dataset contained 155 amygdala (monkey H: 47; monkey L: 108) and 60 vlPFC neurons (monkey H: 39; monkey L: 21) across the 3 drug conditions. Table 1 summarizes all recorded neurons across treatment conditions.

The spiking data was then analyzed using custom written MATLAB scripts (version 2022b; MathWorks, Natick, MA). Instantaneous firing rate for each neuron was extracted and binned into 30 s windows. Two periods were considered and defined as follows: the pre-injection period was from 10 min before and up until the injection timestamp, while the post-injection period started after the injection waiting period (15/30 min for DCZ/CNO) and ended 20 min later. The firing rate for each neuron was normalized using the mean and standard deviation calculated during the pre-injection period (z-score normalization).

At the level of the neuronal populations, changes in firing rate were averaged for both recording period (pre-injection and post-injection) to calculate an average value for each neuron across each period. The difference between these averages (post-injection – pre-injection) was then statistically compared using Wilcoxon signed-rank test. Comparison between drug condition (vehicle vs DCZ or vehicle vs CNO) was performed on this same difference in averages using Kruskal-Wallis tests.

At the level of single neurons, we determined the proportion of neurons modulating their firing rate following drug injection. Neurons were considered modulated if their absolute normalized firing rate during the post-injection period was greater than 1.96 sd for at least 4 time bins (10% of the post-injection period, 2/20 min). To compare the proportion of modulated neurons across drug condition and area, we used mixed-effect logistic regressions, which included area (amygdala, vlPFC), drug condition (vehicle, DCZ or vehicle, CNO), and their interaction as factors, and a random effect of animal (2 levels).

The LFP data were analyzed using the FieldTrip toolbox (20221223 release)[100] and custom-made scripts on MATLAB. First, the wide band signal was subsampled at 1 kHz before applying bipolar-referencing by subtracting the signal of every contact along each electrode array (16 contacts) with its closest neighbor (i.e., 15 bipolar sites). Visual inspection was used to reject bipolar sites containing noise from further analyses (across session average [min-max], 10% [0–20%] of bipolar sites).

To extract robust estimates of power modulation and functional connectivity, we first created 50 non-overlapping bins of 4 s each for both the pre-injection period and post-injection period, as defined above. This trial creation procedure and the following computations were repeated 100 times.

We applied a band-pass filter between 0.5 and 200 Hz and an additional band-stop for line noise at 60 Hz and its harmonics. We then performed a multitaper frequency transformation (7 tapers), with a spectral resolution of 0.5 Hz at each bin (from 1 to 100 Hz) and a spectral smoothing of ±1 Hz. This was used to estimate the power spectrum across areas, time periods and drug conditions. Grand-average power spectrograms in the post-injection period (across the 100 repetitions) were normalized using the average and standard deviation across the 100 pre-injection repetitions at every frequency bin (z-score normalization). Changes in normalized power in the post-injection periods were then assessed at every frequency bin using Wilcoxon signed-rank tests, while drug conditions effects (vehicle vs DCZ or vehicle vs CNO) were tested using Kruskal-Wallis tests. In both cases, we applied a threshold for significance at $p = 0.01$ for 5 consecutive frequency bins. Note that only amygdala recordings ipsilateral to the vlPFC recording sites in monkey L are reported, allowing for a direct comparison with the following coherency analyses.

The multitaper frequency decomposition output was also used to derive the imaginary part of the coherency between every pair of sites across the 50 trials for both time periods. Coherency values at each frequency bin were averaged across the 100 repetitions of the trial creation procedure, resulting in two tensors per recording session (one per time period) of size Nch x Nch x Nfq, where Nch is the number of channels and Nfq the number of frequency bins. For each of the 100 repetitions, we also created a matched null permutation where we scrambled the association between the different bipolar sites by randomizing the trial assignment for each site. This allowed us to create null coherograms. We further focused our analyzed on inter-areal coherence, specifically between ipsilateral amygdala-vlPFC pairs, therefore disregarding the contralateral amygdala in monkey L.

The existence of coherency at each frequency bin was established by comparing it to the null coherogram derived from permutations using Wilcoxon signed-rank tests (with a threshold for significance at $p = 0.01$ for 5 consecutive frequency bins; Supplementary Fig. 5). A frequency band of interest was further defined within the frequency bins showing significant coherencies by extracting the peak frequencies and the full width at a third of the peak maximum across both periods and all drug conditions for each monkey (monkey H: 6.5–14.5 Hz with peak at 10.5 Hz, monkey L: 8–14 Hz with peak at 9 Hz). Given the overlap between monkeys, the frequency band used was based on the minimum and maximum frequencies across both monkeys (6.5–14.5 Hz), encompassing what is usually referred to as alpha oscillations.

Finally, we averaged the coherency observed between a given amygdala site and all possible vlPFC sites, resulting in two vectors of alpha coherency for each recording session, one for each period considered. To test for statistical difference between conditions and monkeys, we used two 2-way ANOVAs which included drug condition (vehicle, DCZ or vehicle, CNO), monkey (H, L), and their interaction as factors.

## Histological processing

Animals were deeply anesthetized with a sodium pentabarbitol solution and perfused transcardially with 0.1 M phosphate buffered saline (PBS) and 1% (w/v) formaldehyde solution derived from depolymerized paraformaldehyde followed by 0.1 M PBS and 4% (w/v) formaldehyde solution derived from depolymerized paraformaldehyde. Brains were removed and immersed in this same formaldehyde solution overnight. Following post-fixation, brains were transferred to a solution of 2% DMSO and 10% glycerol in 0.1 M PBS for 24–48 h, then transferred to a solution of 2% DMSO and 20% glycerol in 0.1 M PBS for at least 24 h. Cryoprotected brains were blocked in the coronal plane, then flash frozen in −80 °C isopentane. After flash freezing, the brains were removed, dried, and stored at −80 °C until sectioning. Tissue was cut serially in 50 μm coronal sections on a sliding microtome (Leica SM 2010R) equipped with a freezing stage.

One series from each brain was mounted on glass slides, Nissl stained using cresyl violet solution, and coverslipped. A second series from each brain was used for confirmation of DREADD expression. Sections through the amygdala were selected and washed with PBS solution containing 0.3% Triton X-100 (TX-100). Endogenous peroxidase was quenched by incubation in 0.6% hydrogen peroxide in PBS for ten min. The sections were washed, then blocked for 1 h in a solution of 1% bovine serum albumin (Cat #BP671-1, Thermo Fisher Scientific, Waltham, MA), 1% normal goat serum (Cat #PI31873, Invitrogen, Waltham, MA), and 0.3% TX-100 in PBS. Primary antibody against the hemagglutinin tag (raised in rabbit, clone C29F4, 1:400, Cell Signaling Technology, Danvers, MA; Cat #3724, RRID:AB_1549585) was added to the blocking solution and sections were incubated overnight. At least one amygdala section for each animal, taken from a third series, was processed in a buffer solution overnight without primary antibody as a control. Incubation in primary antibody was followed by washes and a 2-h incubation in anti-rabbit biotinylated secondary antibodies (1:200, Vector Laboratories, Burlingame, CA, RRID:AB_2313606). Sections were then washed and incubated for 1.5 h in avidin-biotin complex solution (1:200; Vectastain standard kit, Vector Laboratories, RRID:AB_2336819). Finally, sections were washed, incubated in a 3-3'-diaminobenzidine tetrahydrochloride (DAB) solution for 10 min, and 0.006% hydrogen peroxide was added to the solution. Sections were carefully observed for a reaction and were placed in buffer solution after approximately 2.5–3.5 min, when staining was visually apparent. Sections were mounted on glass slides, air dried, dehydrated in ascending concentrations of ethanol, and cleared in d-limonene, then coverslipped with DPX mounting medium.

A fourth series was used to co-stain for Nissl and DREADDs. Finally, a fifth and sixth series were used to co-stain for DREADDs and either inhibitory or excitatory neuronal markers. Aside from the antibodies used, the processing of these series was identical. First, sections through the amygdala were selected and washed with PBS solution containing 0.3% TX-100. The sections were blocked for 1 h in a solution of 5% bovine serum albumin (Cat #PI31873, Thermo Fisher Scientific, Waltham, MA), 10% normal goat serum (Cat #PI31873, Invitrogen, Waltham, MA), 10% normal donkey serum (Cat #D9663-10ML, Sigma-Aldrich, St Louis, MO) and 0.3% TX-100 in PBS. Primary antibody against the hemagglutinin tag (raised in rabbit, clone C29F4, 1:400, Cell Signaling Technology, Danvers, MA; Cat #3724, RRID:AB_1549585), and a second primary antibody against either CaM kinase II, alpha subunit (raised in mouse, clone 6G9, 1: 1:1000, Sigma-Aldrich, St Louis, MO; Cat #05-532, RRID:AB_309787) or GABA (raised in mouse, clone GB-69, 1: 1:100, Sigma-Aldrich, St Louis, MO; Cat# A0310, RRID:AB_476667) was added to the blocking solution and sections were incubated overnight. Incubation in primary antibody was followed by washes and a 2-h incubation, protected from light, in secondary antibody solution. This consisted of donkey anti-rabbit Alexa Fluor 647 conjugated secondary antibody (1:200, Invitrogen, Waltham, MA; Cat # A31573, RRID:AB_2536183) and goat anti-mouse Alexa Fluor 488

conjugated secondary antibody (1:200, Invitrogen, Waltham, MA; Cat # A32723, RRID:AB_2633275). Sections were then washed, mounted on glass slides, coverslipped with ProLong Gold Antifade mountant (Cat # P10144, Invitrogen, Waltham, MA), air dried while protected from light, and stored at −20 °C.

Images of sections were acquired using a Zeiss Apotome 2 microscope equipped with Q-Imaging and Hamamatsu digital cameras, a motorized stage, and Stereo Investigator software (MBF Bioscience; RRID:SCR_002526). Images were acquired using consistent exposure time and brightness settings within the same section. Any adjustments made afterward (e.g., to increase brightness) were applied uniformly to the image. To obtain wide field images of the entire section, the outline of the tissue was defined using a 5x lens. Tiled images of a single section were obtained sequentially, constrained by the boundaries of the tissue, and a 10% overlap and image stitching was used to create a composite image of all tiled photos.

We performed standard unbiased stereological analysis on sections co-stained for Nissl and DREADDs. Using the cell soma as a counting target, cells positive for cresyl violet and HA tag were identified, and numbers of labeled cells were estimated using an optical fractionator probe. Using previous stereological studies in macaque as a guide[101], we chose 5 evenly-spaced sections through the amygdala in each animal. Neurons were counted at 10x magnification, with a counting frame of size $150 \times 150 \times 15$ mm; a 5 mm guard was applied to the dorsal aspect of each section and a 20 mm guard to the ventral side. Counting frames were arranged in a $670.8 \times 670.8$ mm grid for systematic-random sampling. Neurons were distinguished from glia labeled by cresyl violet stain through cell morphology and presence of a well-defined nucleolus. In total, we counted 533 DAB-positive and 10239 Nissl-positive cells in Animal L, and 399 DAB-positive and 7998 Nissl-positive cells in Animal H.

### Reporting summary

Further information on research design is available in the Nature Portfolio Reporting Summary linked to this article.

## Data availability

Source data are provided with this paper. The functional MRI and neurophysiology data generated in this study have been deposited in the Zenodo database under https://doi.org/10.5281/zenodo.10933334 (https://zenodo.org/records/10933335)[102]. Openly available resources used in this study such as the NMT[45], D99[36], CHARM[45], and SARM[44] atlases are accessible through the Primate Data and Resource Exchange (PRIME-DRE; https://prime-re.github.io/). The Mountainsort[99] and Fieldtrip[100] toolboxes are openly available on Github (https://github.com/flatironinstitute/mountainsort5; https://github.com/fieldtrip/fieldtrip). Source data are provided with this paper.

## Code availability

The custom MATLAB scripts used to analyze the functional MRI data are available on Github at https://github.com/RudebeckLab/rsFMRI-dreadds.

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

## Acknowledgements

This work was supported by the following sources of funding: NIMH R01MH110822 (A.F., C.E., F.M.S., N.B., L.L., B.E.R., and P.H.R.); BRAIN Initiative RF1MH117040 (A.F., C.E., F.M.S., N.B., L.L., B.E.R., and P.H.R.); NIMH R01MH111439 (B.E.R.); Takeda Science Foundation Overseas Research Fellowship (A.F.); Brain & Behavior Research Foundation Young Investigator grant #28979 (A.F.); Hope for Depression Research Foundation grant (S.H.F.). We would like to thank Dr Paula Croxson for providing the foundation on which this work was built and Jairo Munoz for assistance with data acquisition. We thank Dr Paul Taylor for help with fMRI data pre-processing. For help with fMRI analysis we thank Drs Alex Franco and Vincent Costa. We also thank Dr Jian Jin for providing clozapine N-oxide powder and Drs Naohisa Miyakawa and Vincent Costa for their advice on setting up S-probe configuration. We thank Dr David Leopold for his advice and suggestions on the manuscript. Finally, we thank the veterinary and animal care staff at Mount Sinai for their expertise and support.

## Author contributions

C.E., A.F., B.E.R. and P.H.R. designed the study. C.E., A.F., S.H.F., L.F. and N.B. collected the neuroimaging data. C.E. and A.F. collected the neural

recording data. C.E. analyzed the neuroimaging data. F.M.S. analyzed the neural recording data. L.L. performed the histological stereology analysis. C.E., A.F., F.M.S., B.E.R. and P.H.R. drafted the manuscript. All authors edited the paper.

## Competing interests

The authors declare no competing interests.
