## [Peer Review File · Nature Communications]

The neural basis of resting-state fMRI functional connectivity in fronto-limbic circuits revealed by chemogenetic manipulationREVIEWER COMMENTS

Reviewer #1 (Remarks to the Author):

In their experiment, Elorette and colleagues investigated the impact of a chemogenetic manipulation of the amygdala. Chemogenetic is a very promising method for manipulating brain activity of selective neuronal population. The presented work is key for establishing the validity of this approach.

By injecting DREADDs bilaterally in the amygdala, the authors showed that inhibiting amygdala activity could result in a modulation of its functional coupling with the rest of the brain. Overall, the authors observed a global increase in functional connectivity. However, the comparison of the fig 3A and 3B suggests that in addition to stronger positive correlations, DCZ also causes stronger anti-correlation with the basal ganglia and the medial motor/premotor cortex. How do the authors interpret this effect?

The authors then explored the electrophysiological basis of the functional connectivity changes. To do so, they focused on recordings done in the amygdala and in a prefrontal cortex region that receive amygdala inputs, the area 12o of the VLPFC. Overall, they observed a significant increased in firing rate in the amygdala following DCZ injection. In the VLPFC, they only noticed a non-significant trend post DCZ injection. The results reported in Fig5C show a larger variance in the firing rate of the VLPFC neurons versus amygdala neurons. Could it reflect that two subpopulations of VLPFC neurons were recorded: neurons that receive projections from the amygdala and VLPFC neurons that do not receive amygdala inputs? If so, do the two subpopulations display a differential impact of DCZ on the LFP coherence between amygdala and VLPFC neurons?

Finally, the authors conducted a crucial analysis by comparing the effect of two ligands: DCZ and CNO.

P24: did the authors mean 2 sessions per monkeys or 4 sessions in total per monkey.

Reviewer #2 (Remarks to the Author):

Elorette / Fujimoto et al present an important study on primate amygdala – prefrontal cortex functional connectivity. This neural connection is target for several neuropsychiatric conditions and the present study is therefore of general interest to clinicians and basic scientists alike. The authors present data from two macaques in which fMRI and electrophysiology from amygdala and ventral prefrontal cortex (vPFC) in combination with DREADD based manipulation of the amygdala has been conducted under anaesthetized conditions. The methods are all state of the art and the paper very well written. In what follows I outline my present concerns:

Major:

1. I would like to understand better which other cortical and subcortical structures are affected by amygdala manipulation, in addition to the examined amygdala – vIPFC connectivity. Could the author please provide more information in the form of a brief summary? On a similar note: Most of the functional imaging maps are hard to interpret, unless one is a expert on monkey neuroanatomy. I would therefore suggest some labels / pointers that help to guide the viewers attention on particular structures of interest.
2. Figure 4 and Figure 3C: I understand that there were for each animal, 2 sessions with each VEH and DCZ. I am not convinced that averaging across 2 sessions makes sense given such a low N. Therefore, please show the connectivity values for the individual sessions.
3. Interpretation that DCZ treatment has no effect on firing rate on vIPFC neurons, but only on alpha coherence. This is an important finding, but it seems to me the authors should also acknowledge the challenge to identify truly connected sites making it a possibility of firing rate changes elsewhere in vIPFC.
4. Understanding well the expressions in Figure 2 is critical for the interpretation of the functional results, including the differences seen in the two animals in some of the results and for the analysis for Figure 5E. It would therefore be advantageous to see a quantification or better characterization of the expression, ideally into excitatory (CamKII?) vs inhibitory (Parvalbumin?) neurons. For Figure 2C, the mentioned labeling of axon terminals in vIPFC is not apparent. In monkey H vIPFC labeling seems to extend to cell bodies (trans-synaptic effect?). In monkey L, vIPFC labeling seems to be absent.

Minor:

1. Use of contrast agent should be explained. MION is commonly used to boost CNR in awake, moving animals. What is the rationale for using it under anaesthetized conditions? MION is known to differentially affect SNR across brain regions. Can the authors be sure that part of the connectivity results are not contaminated by MION SNR effects?

2. Figure 3: Panel A & B Please also show underlying EPI images.
3. Authors mention 'multi-disciplinary' approach, but it seems to me what they are referring to (combining neuroimaging, electrophysiology and DREADD) might be more appropriately called 'multi-method'.
4. Figure 1: Double spacing btw certain letters. It would be good to understand the figure, without the need to go to the main text / consult other figures. Acronyms (VEH, DCZ) should be explained in the legend. Not entirely clear from the figure whether only animal H and not animal L received treatment with VEH and DCZ. I understand CNO was also injected, but this is missing in this figure.
5. I would soften the wording in the discussion to 'aspiration can have off target effects'
6. Figure 5, Electrophysiology: How many neurons in Amygdala show an increase in firing rate? Some neurons also seem to decrease their spiking. Panel D: It is not clear whether the entire raw LFP or a bandpass filtered version was used for the coherence analysis.
7. Supplementary Figure 3A: Is this coherence between amygdala and vIPFC? Same figure, panel B: Is this from one animal or the average of both animals? What is the rationale for focusing on the lower frequencies? Given the increase in amygdala spiking, one would expect a power increase in higher frequency / gamma. Is this the case?
8. Can the authors rule out contributions from anesthesia? Would they expect similar results under non-anesthetized rest conditions (which may be more clinically relevant)?

Reviewer #3 (Remarks to the Author):

I applaud the authors for conducting such an intricate study to examine chemogenetic manipulation of the brain in a nonhuman primate model. Authors use complimentary in vivo techniques of resting-state fMRI and electrophysiology to demonstrate how activation of inhibitory DREADD receptors impact neuronal activity in fronto-limbic circuits. Specifically, the authors find that activation of the inhibitory DREADD receptor in the basolateral amygdala results in increased resting-state functional connectivity and increased local field potential coherency. These findings are paradoxical to what has been previously reported in many rodent studies suggesting that activation of inhibitory DREADD receptors decrease and inhibit neuronal activity. This study and other previous studies highlight the essential species differences between murine models and nonhuman primates and how these chemogenetic techniques are working in each model species. This also demonstrates the importance of nonhuman primate studies for translating these chemogenetic tools for use in human clinical patient populations. Overall, I thought the study was well-designed and conducted, as well as thoroughly described, but I have a few comments that I would like for authors to consider. I have outlined my comments below:

1. The authors attempt to replicate the findings from Grayson, et al, 2016 present a critical finding about the broader use of CNO in past, present, and future chemogenetic studies in rodents and nonhuman primates. Specifically, authors show that using CNO to activate the DREADD receptor results in inconsistent pattern of changes in rs-FC. Taken together the inconsistent changes in rs-FC with CNO activation and that CNO is easily metabolized into clozapine (a highly potent antipsychotic with affinity for multiple receptor types) may suggest that prior work conducted with CNO or low dose clozapine are largely due to the off-target effects of clozapine and not necessarily the DREADD receptor inhibition. In general, I feel that the authors do not emphasize this important point in their discussion of the results.

2. The discussion section lacks any discussion about what these findings mean for behavior. Authors should consider discuss their findings in light of studies in nonhuman primates that have used inhibitory DREADDs in the amygdala (Raper, et al, 2019; Roseboom, et al, 2021). Both studies found decreased behavioral responses (decreased freezing) when activating the inhibitory DREADDs, but both studies are using CNO and/or clozapine as the DREADD ligand of activation. So based on your findings in rs-FC and ephys with DCZ and CNO, are the findings from Raper and Roseboom entirely dependent on clozapine causing a decrease in activity? Would their results have been different if DCZ had been available for DREADD activity and neuronal activity would have increased instead of decreased?

- Raper J, Murphy L, Richardson R, Romm Z, Kovacs-Balint Z, Payne C, Galvan A. Chemogenetic Inhibition of the Amygdala Modulates Emotional Behavior Expression in Infant Rhesus Monkeys. *eNeuro*. 2019 Oct 14;6(5):ENEURO.0360-19.2019. doi: 10.1523/ENEURO.0360-19.2019. PMID: 31541000; PMCID: PMC6791827.

- Roseboom PH, Mueller SAL, Oler JA, Fox AS, Riedel MK, Elam VR, Olsen ME, Gomez JL, Boehm MA, DiFilippo AH, Christian BT, Michaelides M, Kalin NH. Evidence in primates supporting the use of chemogenetics for the treatment of human refractory neuropsychiatric disorders. *Mol Ther*. 2021 Dec 1;29(12):3484-3497. doi: 10.1016/j.ymthe.2021.04.021. Epub 2021 Apr 23. PMID: 33895327; PMCID: PMC8636156.

3. Was the resting-state fMRI conducted prior to chemogenetic transfection surgery? If so, is there any difference in how simply introducing these novel receptors to the brain impact rs-FC?

4. As a minor point, in addition to citing Gomez et al, 2017 regarding the low BBB penetrance of CNO and reverse metabolism to clozapine, authors should also cite Raper et al, 2017 because it was focused on nonhuman primates.

- Raper J, Morrison RD, Daniels JS, Howell L, Bachevalier J, Wichmann T, Galvan A. Metabolism and Distribution of Clozapine-N-oxide: Implications for Nonhuman Primate Chemogenetics. *ACS Chem*

Neurosci. 2017 Jul 19;8(7):1570-1576. doi: 10.1021/acscchemneuro.7b00079. Epub 2017 Mar 30. PMID: 28324647; PMCID: PMC5522181.

Reviewer #4 (Remarks to the Author):

Elorette and colleagues present a comprehensive and wonderfully executed study on the neural basis of resting state functional connectivity revealed by chemogenetics.

First of all I would like to congratulate the team on this study. It is extremely rare to see such a comprehensive account of chemogenetic effects in NHP and the authors have gone above and beyond to describe the effects both from the perspective of fMRI and electrophysiology.

There are however some small aspects about the paper that I wish the authors could address.

On the big picture level I find the framing of the manuscript a little bold to be honest, while the study does give us a more thorough understanding of what happens to functional connectivity in a specialized fronto-limbic circuit when it is perturbed there are no simultaneous recordings that would be needed to give a full account of the relationship between rs-FC and its neuronal substrate. Nevertheless, the framing around the fact that timescales are distinct in frontal circuits and that these distinct connectivity patterns could lead to different “relationships” is interesting and something the reviewer appreciates highly.

One thing worth discussing in the paper is the fact that a non specific DREADDs was used and this likely affects the results. I understand why this approach was chosen but the implications would be worth discussing by the authors.

Overall this is a highly significant paper which is very well written and thoroughly executed. The analysis is rigorous and the results stand on their own. The paper should be accepted with very minor revisions.

Nature Communications – Response to Reviewers

We would like to thank all reviewers for their positive and constructive comments on our manuscript. Below we address each of their concerns in a point-by-point manner. Reviewers' comments are in blue, our responses are in **black** and any changes to the manuscript are highlighted in red.

Reviewer #1:

In their experiment, Elorette and colleagues investigated the impact of a chemogenetic manipulation of the amygdala. Chemogenetic is a very promising method for manipulating brain activity of selective neuronal population. The presented work is key for establishing the validity of this approach.

R1.1: By injecting DREADDs bilaterally in the amygdala, the authors showed that inhibiting amygdala activity could result in a modulation of its functional coupling with the rest of the brain. Overall, the authors observed a global increase in functional connectivity. However, the comparison of the fig 3A and 3B suggests that in addition to stronger positive correlations, DCZ also causes stronger anti-correlation with the basal ganglia and the medial motor/premotor cortex. How do the authors interpret this effect?

Response 1.1: We would like to thank the reviewer for highlighting what we believe is our original poor explanation of the data that is shown in Figure 3. We have updated our description of Figure 3 in the manuscript to more clearly describe the data being shown.

In Figures 3A and 3B we show the amygdala functional connectivity maps before (3A) and after DCZ (3B) is administered. As the reviewer correctly notes there are higher negative correlations between amygdala and basal ganglia as well premotor cortex in both of these figures. However, the key comparison is in Figure 3C which isolates the effect of administering DCZ by showing the difference between DCZ administration and vehicle administration. In other words, Figure 3C shows the change in DCZ over and above the vehicle control condition as it shows the subtraction between the DCZ post-injection minus pre-injection data and the vehicle post-injection minus pre-injection data. As can be seen in this figure, there was a slight increase in functional connectivity between amygdala and premotor cortex, and there was a slight decrease in functional connectivity between amygdala and basal ganglia. As Reviewer 2 also requested more information about changes in regions other than ventrolateral PFC (vIPFC), we have expanded our interpretation of the results to highlight additional effects outside the amygdala-PFC network that we had originally focused on. The following changes have been made to the manuscript:

Results, page 8:

“The results of **the subtraction of rs-FC due to vehicle from rs-FC due to DCZ** for one animal are shown in **Figure 3D** (more examples shown in **Supplemental Figure 3**). **We**

observed consistent effects across both animals in regions where rs-FC was increased after DREADD activation with DCZ as compared to vehicle (e.g. anterior cingulate cortex, orbitofrontal cortex, ventrolateral prefrontal cortex, insula, premotor cortex, hippocampus, superior temporal gyrus) as well as regions where rs-FC was decreased (globus pallidus, caudate tail, inferior frontal gyrus, middle temporal gyrus, V1, V2). Overall, we found that there was a global change in rs-FC between the amygdala and the rest of the brain ($p < 0.0001$, **Figure 3E**).“

R1.2: The authors then explored the electrophysiological basis of the functional connectivity changes. To do so, they focused on recordings done in the amygdala and in a prefrontal cortex region that receive amygdala inputs, the area 12o of the VLPFC. Overall, they observed a significant increased in firing rate in the amygdala following DCZ injection. In the VLPFC, they only noticed a non-significant trend post DCZ injection. The results reported in Fig5C show a larger variance in the firing rate of the VLPFC neurons versus amygdala neurons. Could it reflect that two subpopulations of VLPFC neurons were recorded: neurons that receive projections from the amygdala and VLPFC neurons that do not receive amygdala inputs? If so, do the two subpopulations display a differential impact of DCZ on the LFP coherence between amygdala and VLPFC neurons?

Response 1.2: We agree with the reviewer’s thoughtful suggestion that the variance in the vIPFC neurons is worth further investigation. We are not able to determine if the vIPFC neurons were recorded from were also neurons expressing DREADDs, and the relationship between local excitability in vIPFC and the strength and extent of DREADD-expressing neurons from the amygdala is difficult to assess. As a way to answer the question of how vIPFC spiking might relate to coherency, we reanalyzed the coherency between vIPFC and AMY following DCZ injection depending on whether vIPFC neurons recorded on a given channel increased or decreased their firing rate. For this, we considered all recorded vIPFC neurons and found that 8 showed a relative increase in firing rate following DCZ administration, while 8 showed a decrease (2 more increasing neurons were not considered due to noise in the LFP channel). We then averaged the coherency observed between all recorded AMY bipolar derivations and the bipolar sites where each of these vIPFC neurons were recorded (e.g. if a neuron was recorded on channel 6, we considered the bipolar derivations between channel 5-6 and 6-7). Overall, we did not observe differences in coherency between AMY and vIPFC depending on whether vIPFC neurons increased or decreased firing rate (Kruskal-Wallis test, $\text{Chi}^2=0.47$, $p=0.49$, see new Supplemental Figure 4C reproduced below).

However, given that our population of neurons recorded in vIPFC was sparse, we are not powered to fully answer the question of how sub-populations of neurons might have contributed to overall changes in coherency between these regions. We have added the following text to the results, page 13, and the discussion section, page 19, to expand on this point:

Results (page 13):

“Finally, because single neuron firing rate changes in vIPFC were variable, we ran a post-hoc analysis to see if the direction of change in firing rate was related to changes in LFP coherency between amygdala and vIPFC on those same bipolar sites. We found that there was no difference in amygdala to vIPFC coherence when vIPFC channels were separated by whether neurons on those channels either increased or decreased their firing rates (**Supplemental Figure 4C**, Kruskal-Wallis test, $\chi^2=0.47$, $p=0.49$). This analysis reveals that there is not a tight relationship between changes in firing rate and local changes in coherence.”

Discussion (page 20):

“We observed that activating inhibitory DREADDs in the amygdala caused an increase in neural spiking locally, but not in vIPFC. It is important to note that previous chemogenetic studies in macaques, including our own, were not able to determine if the recorded neurons are those that have expressed DREADD receptors, or if they are simply communicating with DREADD-expressing neurons. **Therefore, our neurophysiological sampling in the amygdala does not necessarily represent the firing changes solely in DREADD-expressing neurons.** Further, when recording from a site downstream of the target region, as we performed here with vIPFC, it is difficult to know how the extent and strength of the projections will affect excitability. In our data, we did not observe a relationship between changes in neural spiking in the vIPFC and changes in amygdala-vIPFC LFP coherence. We therefore consider our findings from the perspective of modulations in overall local excitability, as opposed to direct measures of individually affected neurons.”

We have also added the following figure as a panel as Supplemental Figure 4C:

**DCZ-induced change in
Amygdala-vIPFC LFP coherency
by vIPFC firing rate change**

Supplemental Figure 4C. Average changes in amygdala-vIPFC LFP coherency after DREADD inhibition with DCZ, grouped by changes in local firing rate in vIPFC. Grey dots indicate average coherency values between vIPFC bipolar sites where neurons were recorded and all amygdala bipolar sites. vIPFC bipolar sites where we recorded neurons that increased their firing rate after DCZ injection are shown on the left, dark green box plot; vIPFC bipolar sites where we recorded neurons that decreased their firing rate after DCZ injection are shown on the right, light green box plot.

R1.3: Finally, the authors conducted a crucial analysis by comparing the effect of two ligands: DCZ and CNO. P24: did the authors mean 2 sessions per monkeys or 4 sessions in total per monkey.

Response 1.3: We have clarified the language to describe the experimental design. This section now reads:

Methods, page 25:

“Each animal completed five imaging sessions in total. Both monkeys H and L completed two sessions each of imaging with vehicle and DCZ injection. Drug sessions were alternated and the order of DCZ and vehicle testing was counterbalanced across animals. After the conclusion of the vehicle and DCZ sessions, both animals completed one session where CNO was administered.”

Reviewer #2:

Elouette / Fujimoto et al present an important study on primate amygdala-prefrontal cortex functional connectivity. This neural connection is target for several

neuropsychiatric conditions and the present study is therefore of general interest to clinicians and basic scientists alike. The authors present data from two macaques in which fMRI and electrophysiology from amygdala and ventral prefrontal cortex (vPFC) in combination with DREADD based manipulation of the amygdala has been conducted under anaesthetized conditions. The methods are all state of the art and the paper very well written. In what follows I outline my present concerns:

We appreciate the reviewer's thoughtful feedback. We address each of their points below.

Major:

R2.1: I would like to understand better which other cortical and subcortical structures are affected by amygdala manipulation, in addition to the examined amygdala-vPFC connectivity. Could the author please provide more information in the form of a brief summary? On a similar note: Most of the functional imaging maps are hard to interpret, unless one is an expert on monkey neuroanatomy. I would therefore suggest some labels / pointers that help to guide the viewers attention on particular structures of interest.

Response 2.1: Reviewer 1 made a similar comment about providing more information about the full pattern of effects. We are happy to provide more information on this point and have made the following changes to the manuscript:

Results, page 8:

"The results of the subtraction of rs-FC due to vehicle from rs-FC due to DCZ for one animal are shown in **Figure 3D** (more examples shown in **Supplemental Figure 3**). We observed consistent effects across both animals in regions where rs-FC was increased after DREADD activation with DCZ as compared to vehicle (e.g. anterior cingulate cortex, orbitofrontal cortex, ventrolateral prefrontal cortex, insula, premotor cortex, hippocampus, superior temporal gyrus) as well as regions where rs-FC was decreased (globus pallidus, caudate tail, inferior frontal gyrus, middle temporal gyrus, V1, V2). Overall, we found that there was a global change in rs-FC between the amygdala and the rest of the brain ($p < 0.0001$, **Figure 3E**)."

We have also updated Figure 3 to include labels to identify regions around the amygdala and vPFC, as well as highlighted these regions in one hemisphere.

Figure 3. Representative changes in functional connectivity between amygdala and frontal cortex. A) Coronal view of the standardized macaque atlas showing vIPFC (left) and amygdala (right) ROIs highlighted on the left hemisphere. Select anatomical regions are labeled on the right hemisphere. **B-D)** Average rs-FC with a left amygdala seed region in animal L. Pre-DCZ injection period (**B**) and post-injection period (**C**). Scale bar indicates z-score of rs-FC. Threshold $p=0.0028$, cluster size ≥ 30 voxels, voxel faces touching, average of two sessions. **D)** Difference in FC produced by DREADD inhibition [(DCZ post-injection – pre-injection) – (VEH post-injection – pre-injection)], calculated from averaged DCZ post-injection – pre-injection data and averaged VEH post-injection – pre-injection data. Scale bar indicates difference in z-score of rs-FC. Threshold $p=0.0485$, cluster size ≥ 30 voxels, voxel faces touching. **E)** Average change in rs-FC between amygdala (bilateral ROI) and all other voxels in the brain after treatment with vehicle or DREADD activation via DCZ. Symbols indicate individual session values for each animal. Animal H, diamond; animal L, plus sign.

Multiway ANOVA, main effect of drug $F[1, 226780] = 272.20$, $p < 0.0001$, main effect of animal $F[1, 226780] = 186.82$, $p < 0.0001$, interaction of drug and animal $F[1, 226780] = 64.26$, $p < 0.0001$. *ACC = anterior cingulate cortex; OFC = orbitofrontal cortex; vIPFC = ventrolateral prefrontal cortex; Str = striatum; AMY = amygdala; PrMo = premotor cortex; Ins = insula; DCZ = deschloroclozapine; VEH = vehicle.*

R2.2. Figure 4 and Figure 3C: I understand that there were for each animal, 2 sessions with each VEH and DCZ. I am not convinced that averaging across 2 sessions makes sense given such a low N. Therefore, please show the connectivity values for the individual sessions.

Response 2.2. We are happy to show the values for the individual animal sessions and have updated Figures 3D (now 3E; see Response 2.1) and 4 to include points for each session along with the group average and variance (bar graph). We are not able to alter Figure 3C (now 3D) in this way, as this panel shows the contrast between the DCZ and VEH treatment; this comparison requires subtracting data across sessions, and there is no a priori reason to only use two of the four sessions.

Although common in the field of NHP research, we acknowledge the reviewer's concern that we only studied two animals and that this is fewer than would be used in comparable studies in humans. However, the power in our study design comes instead from other features, specifically the extended within session data and replication within-subject. Previous work has demonstrated that greater than 30mins of resting-state data in NHP provides a high degree of reliability within an individual subject (Xu et al. 2018; Autio et al. 2018; Yacoub et al. 2020), however there is some variation that occurs from session to session. For this reason, we chose to collect pre- and post-injection data for our treatments to reduce the influence of across session variability. Additionally, collecting replications help to further improve our signal to noise within subjects (Yacoub et al. 2020; Areshenkoff et al. 2021). Outside of acquisition, analysis of our results with an ROI defined a priori and comparing our findings to well characterized resting state networks helps to reduce possible errors due to multiple comparisons and improve our power. For all these reasons, we are confident in the strength of our findings and feel that averaging across session, within-subject, helps to improve our signal to noise. However, as stated above, where appropriate we have added individual sessions results to the figures and manuscript (see below).

We have also updated the **methods**, page 25, to reflect the discussion above:

“The power in this study design is derived from the extended within-session acquisition as well as from the replications within-subject. Previous work has demonstrated that greater than 30 minutes of resting-state data in NHPs provides a high degree of reliability within an individual subject (Xu et al., 2018; Yacoub et al., 2020; Autio et al., 2021), however, some variation occurs across sessions. The collection of pre- and post-injection data for each treatment reduces the influence of across-session variability. Collecting replications improves the signal to noise ratio within subjects (Yacoub et al., 2020; Areshenkoff et al., 2021). To reduce multiple comparisons and improve power

outside of data acquisition, fMRI results were analyzed using *a priori* defined ROIs and results were compared to well characterized resting state networks.”

Figure 4. Whole brain fMRI functional connectome altered by chemogenetic inhibition of amygdala. Average FC across all ROIs calculated across a standard whole brain atlas (A), a cortical atlas (B) and a subcortical (C) atlas. Activation of DREADDs via DCZ increased global rs-FC across all three networks. Symbols denote average difference in rs-FC for each animal, separated by session. Animal H, diamond; animal L, plus sign. Error bars represent SEM. Atlas image insets shown on NMT v2 (Seidnitz et al. 2018, Hartig et al. 2021, Jung et al. 2021, Saleem et al. 2021). Multi-way ANOVA analyses: Whole brain, main effect of drug $F[1, 146884] = 728.35, p < 0.0001$ and subject $F[1, 146884] = 498.91, p < 0.0001$, interaction between subject and drug $F[1, 146884] = 12.54, p = 0.0004$. Cortical atlas, main effect of drug $F[1, 5036] = 34.71, p < 0.0001$ and subject $F[1, 5036] = 115.59, p < 0.0001$. Subcortical atlas, main effect of drug $F[1, 4484] = 44.61, p < 0.0001$ and subject $F[1, 4484] = 56.76, p < 0.0001$, interaction between subject and drug $F[1, 4484] = 206.47, p < 0.0001$. DCZ = deschloroclozapine; VEH = vehicle.

R2.3. Interpretation that DCZ treatment has no effect on firing rate on vIPFC neurons, but only on alpha coherence. This is an important finding, but it seems to me the authors should also acknowledge the challenge to identify truly connected sites making it a possibility of firing rate changes elsewhere in vIPFC.

Response 2.3: We thank the reviewer for highlighting this important point. The following changes have been made to the **discussion**, page 20:

“We observed that activating inhibitory DREADDs in the amygdala caused an increase in neural spiking locally, but not in vIPFC. It is important to note that previous chemogenetic studies in macaques, including our own, were not able to determine if the recorded neurons are those that have expressed DREADD receptors, or if they are simply communicating with DREADD-expressing neurons. Therefore, our neurophysiological sampling in the amygdala does not necessarily represent the firing changes solely in DREADD-expressing neurons. Further, when recording from a site downstream of the target region, as we performed here with vIPFC, it is difficult to know how the size and strength of the projections will affect excitability. In our data, we did not observe a relationship between changes in neural spiking in the vIPFC and local changes in amygdala-vIPFC LFP coherence. We might therefore consider our findings from the perspective of modulations in overall local excitability, as opposed to direct measures of individually affected neurons.”

R2.4. Understanding well the expressions in Figure 2 is critical for the interpretation of the functional results, including the differences seen in the two animals in some of the results and for the analysis for Figure 5E. It would therefore be advantageous to see a quantification or better characterization of the expression, ideally into excitatory (CamKII?) vs inhibitory (Parvalbumin?) neurons. For Figure 2C, the mentioned labeling of axon terminals in vIPFC is not apparent. In monkey H vIPFC labeling seems to extend to cell bodies (trans-synaptic effect?). In monkey L, vIPFC labeling seems to be absent.

Response 2.4: We agree with the reviewer that providing a better anatomical account of the functional effects would strengthen our manuscript. To better show the distribution of the DREADDs expression, we have altered Figure 2 to include images at higher magnification (40x) for each animal. These new images now clearly show the expression of DREADD receptors in the axon terminals in vIPFC.

Additionally, in response to this feedback we performed stereology of the amygdala across five sections per animal in a set of sections that had been co-stained for HA and Nissl. Although Nissl is not specific for neurons, we attempted to count only neurons based on morphology of the cells and presence of nuclear staining (see Upright et al. *Journal of Neuroscience*, 2018). Based on these results, our approach labeled at a minimum 5% of cells in the amygdala. This finding was true for both animals (Animal H 4.99% cells positive for both HA and Nissl stain compared to just Nissl stain; Animal L 5.21%). We have added these quantifications to the **results** section, page 6:

“Immunohistochemical analysis targeted to the HA marker protein revealed robust expression throughout the amygdala (**Figure 2A**). We performed stereological analysis of the amygdala in both animals, using sections double-stained for the HA marker protein and Nissl. We observed DREADD labeling of approximately 5% of cells in the amygdala. This finding was true for both animals (Animal H 4.99% cells positive for both HA and Nissl stain compared to just Nissl stain; Animal L 5.21%). In both animals, the strongest labeling occurred in basal and basal accessory nuclei.”

We have added a description of the stereology analysis to the **methods** section as follows on page 31:

“We performed standard unbiased stereological analysis on sections co-stained for Nissl and DREADDs. Using the cell soma as a counting target, cells positive for cresyl violet and HA tag were identified, and numbers of labeled cells were estimated using an optical fractionator probe. Using previous stereological studies in macaque amygdala as a guide (Upright et al., 2018; Zeisler et al., 2023), we chose 5 evenly-spaced sections through the amygdala in each animal. Neurons were counted at 10x magnification, with a counting frame of size 150 x 150 x 15 mm; a 5-mm guard was applied to the dorsal aspect of each section and a 20-mm guard to the ventral side. Counting frames were arranged in a 670.8 x 670.8 mm grid for systematic-random sampling. Neurons were distinguished from glia labeled by cresyl violet stain through cell morphology and presence of a well-defined nucleolus. In total, we counted 533 DAB-positive and 10239 Nissl-positive cells in Animal L, and 399 DAB-positive and 7998 Nissl-positive cells in Animal H.”

We also performed an analysis of inhibitory and excitatory neuron co-localization with DREADD receptors. Here we display at least one section each per animal that we stained for HA (red) and either CaMKII (green, a marker of excitatory neurons) or GABA (green, a marker of inhibitory neurons). Across both treatments, we observed a greater colocalization of neurons with DREADD receptors and either excitatory or inhibitory markers close to the injection sites. Our qualitative analysis did not reveal a difference in proportion of dual labeled neurons between the excitatory and inhibitory stains; we therefore believe that it is unlikely that inhibitory DREADD receptors were preferentially expressed by inhibitory or excitatory neurons. However, should the reviewer feel strongly, we are happy to include a full stereological analysis of inhibitory and excitatory marker overlap with DREADD receptors but note that this is a long and extensive analysis that would take months to complete. We chose not to include such an analysis here as based on the above, we do not believe this would result in a significant difference. This is, however, an important question so we have now included the following text and additional figure in the manuscript, as the new **Supplemental Figure 1**:

Supplemental Figure 1. Colocalization of DREADD receptors with markers for excitatory or inhibitory neurons in the amygdala. Left amygdala sections shown for animals H (left, A and C) and L (right, B and D) stained using an immunofluorescent approach. Images were taken at 5x magnification in a tiled arrangement and stitched together. Insets, top left, were taken at 10x magnification from the same section. Two white chevrons in each inset show examples of dual-labeled cells. **A-B.)** Sections stained for the DREADD receptor's HA tag (red) and CaMKII (green). Dual labeled cells appear yellow. **C-D.)** Sections stained for the DREADD receptor's HA tag (red) and GABA (green). Dual labeled cells appear yellow.

We also added the following text to describe the **results** shown in the new Supplemental Figure 1, page 6:

“When sections throughout the amygdala were co-stained for DREADD receptors and inhibitory or excitatory neuronal markers, we found that inhibitory DREADD receptors were expressed in both types of neurons across the amygdala (**Supplemental Figure 1**). Further, labeled axon terminals were also visible in both animals throughout the amygdala, as well as in regions known to receive projections from basal amygdala, such as the ventral frontal cortex (**Figure 2C**).”

We have updated the **methods** to describe this additional staining and tissue processing as follows on page 30:

“A fourth series was used to co-stain for Nissl and DREADDs. Finally, a fifth and sixth series were used to co-stain for DREADDs and either inhibitory or excitatory neuronal markers. Aside from the antibodies used, the processing of these series was identical. First, sections through the amygdala were selected and washed with PBS solution containing 0.3% TX-100. The sections were blocked for 1 hour in a solution of 5% bovine serum albumin (Cat #BP671-1, Thermo Fisher Scientific, Waltham, MA), 10% normal goat serum (Cat #PI31873, Invitrogen, Waltham, MA), 10% normal donkey serum (Cat #D9663-10ML, Sigma-Aldrich, St Louis, MO) and 0.3% TX-100 in PBS. Primary antibody against the hemagglutinin tag (raised in rabbit, clone C29F4, 1:400, Cell Signaling Technology, Danvers, MA; Cat #3724, RRID:AB_1549585), and a second primary antibody against either CaM kinase II, alpha subunit (raised in mouse, clone 6G9, 1: 1:1000, Sigma-Aldrich, St Louis, MO; Cat #05-532, RRID:AB_309787) or GABA (raised in mouse, clone GB-69, 1: 1:100, Sigma-Aldrich, St Louis, MO; Cat# A0310, RRID:AB_476667) was added to the blocking solution and sections were incubated overnight. Incubation in primary antibody was followed by washes and a 2-hour incubation, protected from light, in secondary antibody solution. This consisted of donkey anti-rabbit Alexa Fluor 647 conjugated secondary antibody (1:200, Invitrogen, Waltham, MA; Cat # A31573, RRID:AB_2536183) and goat anti-mouse Alexa Fluor 488 conjugated secondary antibody (1:200, Invitrogen, Waltham, MA; Cat # A32723, RRID:AB_2633275). Sections were then washed, mounted on glass slides, coverslipped with ProLong Gold Antifade mountant (Cat # P10144, Invitrogen, Waltham, MA), air dried while protected from light, and stored at -20°C.

Images of sections were acquired using a Zeiss Apotome 2 microscope equipped with Q-Imaging and Hamamatsu digital cameras, a motorized stage, and Stereo Investigator software (MBF Bioscience; RRID:SCR_002526).”

We also made the following changes to the **discussion** as the relatively even distribution of DREADD receptors across inhibitory and excitatory neurons has implications for the explanation of the finding that activating inhibitory chemogenetic receptors led to an increase in firing. The following change has been made the manuscript on page 20:

“Only three recent studies in macaques have recorded spiking activity from neurons after DREADD transfection in the target region (Deffains et al., 2021; Hasegawa et al., 2022; Perez et al., 2022), and all observed a decrease in firing rate after chemogenetic inhibition of the target. However, these studies made use of constructs that targeted various populations of cells (respectively, all neurons, all cells, and all catecholaminergic neurons), none recorded from amygdala, and none of these targeted fronto-limbic circuitry. Our own approach targeted all neurons, and so our transfected cells included both inhibitory and excitatory neurons in the amygdala (**Supplemental Figure 1**). Although this makes our inhibitory chemogenetic approach more comparable

to the effects of pharmacological inhibition or excitotoxic lesion, it does alter the interpretation of the effects on fMRI and neurophysiology. To this point, our finding that activating an inhibitory DREADD in macaque amygdala led to increased spiking in this area (**Figure 5**) could be related to the composition of this structure in primates. Recent studies have shown that there is an increased diversity of interneurons in macaque amygdala compared to rodents (Beyeler and Dabrowska, 2020; McDonald and Augustine, 2020; McDonald, 2021). Thus, we speculate that inhibiting these cells may have caused a paradoxical disinhibition of local amygdala circuits, which led to increased extracellular activity. A second possibility is that the inhibition of local excitatory neurons that drive inhibitory networks within the amygdala could have led to increased local activity (Smith et al., 2000). These explanations are not mutually exclusive, and future work may serve to clarify the mechanisms at play. **The use of DREADD constructs specific for excitatory neurons, or DREADD constructs that can target specific projections between two areas, would be a valuable next step.**"

R2.M1. Use of contrast agent should be explained. MION is commonly used to boost CNR in awake, moving animals. What is the rationale for using it under anaesthetized conditions? MION is known to differentially affect SNR across brain regions. Can the authors be sure that part of the connectivity results are not contaminated by MION SNR effects?

Response 2.M1: To the reviewer's point about the use of MION in awake vs anesthetized fMRI procedures, we note that early reports using MION as an MR contrast agent were conducted in studies of anesthetized rodents (Mandeville et al. 1998). More recently, MION has been used in several other anesthetized resting state studies in macaques. Grayson et al. 2016 has a particularly extensive set of analyses showing that MION improves SNR during anesthetized rs-fMRI as compared to BOLD, and Xu et al. 2018 provides evidence that MION improves rs-FC data quality in both awake and anesthetized states. These results, among others, suggest that MION is appropriate and effective for use in anesthetized rs-fMRI (Autio et al 2021).

We expect that any region-specific changes caused in SNR by MION treatment should not influence our results as animals received the same dosage of MION in both vehicle and DCZ or CNO treatments. If MION affected vIPFC differentially from other regions, this effect should remain the same across sessions, and therefore any changes we observed should be controlled for by our study design. To address this concern we have made the following changes to the **methods** on page 25:

"All of the anesthetized neuroimaging conducted here was done using the contrast agent MION, as a number of prior studies in macaques has shown that it improves the quality of rs-FC data in both awake and anesthetized states (Vanduffel et al., 2001; Leite et al., 2002; Grayson et al., 2016; Xu et al., 2019)."

R2.M2. Figure 3: Panel A & B Please also show underlying EPI images.

Response R2.M2. We appreciate the reviewers desire to see the raw underlying EPI data from which our results are derived. We agree that an examination of the raw data provides a better understanding of the limitations of any derived results. Based on the reviewer's suggestion we have included a set of EPI images, however we have not placed them as the underlays in Figure 3 as we wish for the underlying anatomy of the regions to be clear. Instead, we have expanded Supplemental Figure 1 (now Supplemental Figure 2) to show the average value across the time course of the EPI images after the blur step of the preprocessing but before the images are scaled to percent change. These EPIs are taken from the same animal shown in Figure 3. This allows the reader to find and investigate the raw underlying data, while also providing clear anatomical landmarks for interpretation of the results.

Supplemental Figure 2. Representative changes across a single imaging session before and after injection of DCZ. A-B) Changes in functional connectivity between amygdala and frontal cortex, unthresholded connectivity maps. Changes in FC with a left amygdala seed region in animal L in a single session, before and after DREADD activation with DCZ. Pre-DCZ injection period (A) and post-injection period (B). Scale bar indicates z-score of rs-FC. No threshold or clustering has been applied. C-D) Average EPI values across the time series for the same session as in A-B. Pre-DCZ injection period, average across three EPI runs (C), and post-injection period, average across three EPI runs (D). Values were calculated after the blurring step of the preprocessing.

R2.M3. Authors mention “multi-disciplinary” approach, but it seems to me what they are referring to (combining neuroimaging, electrophysiology and DREADD) might be more appropriately called “multi-method”.

Response R2.M3: We used the term “multi-disciplinary” as the different approaches of neuroimaging, neurophysiology, and chemogenetics are so often used in isolation. For instance, neuroimaging investigations in humans are often conducted alone. This is why we used this terminology. The reviewer is correct that it might be more objective to use the term “multi-modal” as this refers to the multiple techniques being used. Given this, we are happy to make the changes to the manuscript as follows:

Abstract: “Thus, our **multi-modal** approach reveals the specific signature of neuronal activity that underlies rs-FC in fronto-limbic circuits.”

Introduction, page 4: “Here we set out to determine the basis of rs-FC in fronto-limbic circuits using a **multi-modal** approach wherein we causally manipulated neural activity using Designer Receptors Exclusively Activated by Designer Drugs (DREADDs, **Figure 1**; Armbruster et al., 2007).”

R2.M4. Figure 1: Double spacing btw certain letters. It would be good to understand the figure, without the need to go to the main text / consult other figures. Acronyms (VEH, DCZ) should be explained in the legend. Not entirely clear from the figure whether only animal H and not animal L received treatment with VEH and DCZ. I understand CNO was also injected, but this is missing in this figure.

Response R2.M4: The reviewers point about readability is well taken. We have remade Figure 1 to reflect the suggested changes. The figure legend has been updated with the following changes:

Figure 1. Schematic of experimental design and analysis approach. A) Experimental timeline from injection of DREADDs into amygdala, through data collection of fMRI and extracellular recordings. **VEH = vehicle; DCZ = deschloroclozapine; CNO = clozapine-N-oxide.** Types of B) fMRI or C) neural activity analyses performed.

R2.M5. I would soften the wording in the discussion to ???aspiration can have off target effects???

Response R2.M5: We have made this change. The sentence in the **discussion** on page 19 now reads:

Aspiration lesions **can** have off-target effects (Meunier et al., 1999; Rudebeck et al., 2013b), further limiting the insights that can be gained from this approach.

R2.M6. Figure 5, Electrophysiology: How many neurons in Amygdala show an increase in firing rate? Some neurons also seem to decrease their spiking. Panel D: It is not clear whether the entire raw LFP or a bandpass filtered version was used for the coherence analysis.

Response R2.M6: Reviewer 1 also had questions about the variability of the electrophysiological responses (R1.2). Based on these comments, we have expanded our description of the electrophysiology data to more precisely catalog the changes that

we observed, and revised the results section for the electrophysiology data. While on average there was an increase in activity across the population of neurons in amygdala, we did indeed find that some neurons that reduced their firing rate after administration of DCZ. Further, we observed some variability in both amygdala and vIPFC across drug treatments. To acknowledge this, we have added the following text to summarize the changes across regions and treatments:

Results, page 12:

“We found that activating the inhibitory DREADD receptors with DCZ increased the firing rate of the recorded amygdala neurons as compared to **the vehicle** treatment (**Figure 5B**). Additionally, only amygdala neurons after DCZ treatment altered their firing rate significantly as compared to the pre-injection period. In the vIPFC, DCZ did not alter firing rate as compared to **vehicle**, and neither treatment altered firing rate significantly from the pre-injection period (**Figure 5C**; **see methods for details on firing rate modulation analysis**). **Specifically, we recorded 46 neurons in the amygdala in the vehicle condition. Of these, 12 (~26%) were considered significantly modulated with an increase in firing rate. Conversely, 5 (~11%) decreased their firing rate. After DCZ treatment, we observed that of the 53 neurons recorded from the amygdala, 32 (~60%) increased, and 6 (~11%) decreased their firing. In the vIPFC, we recorded from 27 neurons in the vehicle condition, and observed 7 (~26%) that increased, and 3 (~11%) that decreased their firing rate. Of the 18 vIPFC neurons recorded in the DCZ condition, 9 (50%) increased and 3 (~17%) decreased their firing.**”

Additionally, we now include a new **Supplemental Figure 4**, which shows the activity of individual neurons across the pre- and post-injection time periods, normalized to the average firing rate in the pre-injection period.

Results, page 12:

“First, we analyzed neuronal spiking activity in both recorded areas to establish whether the observed rs-FC changes were directly associated with changes in firing rate. We recorded the activity of 99 single neurons in amygdala across the pre- and post-injection periods for DCZ or VEH treatment (see **Table 1**; **Supplemental Figure 4**).”

Supplemental Figure 4. Firing rate changes across time after activation of DREADDs with DCZ or after treatment with vehicle. Change in firing rate of each recorded neuron in the amygdala (A) and vIPFC (B), normalized to the average pre-injection firing rate, shown in thirty second bins. Data are separated by treatment (vehicle, top, or DCZ, bottom) and by recording period (pre-injection, left, 0-15 minutes, and post-injection, right, 0-20 minutes). Cool colors indicate a negative change in firing rate from the pre-injection period, warm colors indicate a positive change. C) Average changes in amygdala-vIPFC LFP coherency after DREADD inhibition with DCZ, grouped by changes in local firing rate in vIPFC. Grey dots indicate average coherency values between vIPFC bipolar sites where neurons were recorded and all amygdala bipolar sites. vIPFC bipolar sites where we recorded neurons that

increased their firing rate after DCZ injection are shown on the left, dark green box plot; vIPFC bipolar sites where we recorded neurons that decreased their firing rate after DCZ injection are shown on the right, light green box plot.

Finally, we appreciate the reviewer's confusion about the details of the analysis approach that we employed, and we have updated the legend for Figure 5 to provide more details on the LFP signal and analysis as follows:

D) Average (\pm SEM) change in LFP coherency at each amygdala bipolar site with all vIPFC bipolar sites. Dots indicate amygdala bipolar sites. LFP signal at each bipolar site was band-pass filtered between 0.5-200Hz before a multi-taper frequency transformation was applied; see Methods.

R2.M7. Supplementary Figure 3A: Is this coherence between amygdala and vIPFC? Same figure, panel B: Is this from one animal or the average of both animals? What is the rationale for focusing on the lower frequencies? Given the increase in amygdala spiking, one would expect a power increase in higher frequency / gamma. Is this the case?

Response R2.M7: The reviewer is correct that in previous Supplemental Figure 3A (now Supplemental Figure 5A) we are showing the coherence between amygdala and vIPFC. We agree that our legend was not clear enough and we have updated the figure and legend to more clearly specify this relationship. Similarly, we have clarified the figure and legend to more clearly state what data is being shown in now Supplemental Figure 5B. The legend for Supplemental Figure 5B now also specifies that the figure shows the average power from both animals.

The analysis shown in Supplemental Figure 3B showed changes in power, normalized to the baseline, only in the frequency band in which we observed changes in coherency. We chose to present the data in this way to demonstrate that the change in coherency in the alpha band is not driven by a change in power, and therefore is likely to be driven by changes in phase consistency. After revising Supplemental Figure 3B, now Supplemental Figure 5B, we include the full range of power changes after treatment with DCZ or vehicle, and highlight the frequency band of interest in a cutout at the top right of the panel. Colored bars at the bottom of the figure indicate where post-injection power differs significantly from pre-injection power. The black lines indicate a significant difference between DCZ and vehicle treatment, our comparison of interest. There were differences across the power spectrum between the post-injection and pre-injection periods for both treatments, in both areas, but we observed a significant decrease after DCZ treatment compared to vehicle treatment only in the amygdala, in the beta/gamma range.

We have made the following changes to the figure and figure legend:

Supplemental Figure 5. Across the alpha band, coherency between amygdala and vIPFC is increased, but power is unaltered following DCZ treatment compared to vehicle. **A)** Average coherency between amygdala and vIPFC bipolar sites across all periods and drug conditions for both animals. Bottom lines indicate difference in coherency from randomized permutations (dotted line). Lines/circles (top) show the peak frequency + 1/3rd of the max value. **B)** Post-injection power normalized to the pre-injection period, for the amygdala (top) and vIPFC (bottom) after treatment with DCZ or VEH. Inset (top right corner, magnified from box) shows the power across the frequency band of interest defined in A. Power is averaged across both animals. Dotted line (bottom of graph; blue for VEH treatment, green for DCZ treatment) shows significant differences from pre-injection power (Wilcoxon signed rank test, $p < 0.01$ for 5 consecutive bins). Black dotted line (bottom of graph) indicates significant difference between DCZ and VEH treatments. We observed a decrease in power in the beta-gamma range (20-60 Hz) after DCZ compared to VEH treatment. No significant differences between DCZ and VEH were observed in either region across the alpha band, where coherency was altered (Kruskal-Wallis test, $p < 0.01$ for 5 consecutive bins).

We have revised the **results** section, page 13:

“Next, we looked at changes in LFPs. Here, we first looked at how activating the DREADD receptors altered the power oscillations in the LFP in both amygdala and vIPFC. Based on the prior analysis of single neuron activity, we might expect that increased neural spiking in amygdala would be associated with increased power in higher frequency bands. Indeed, we observed a statistically significant increase in power after DCZ treatment compared to baseline only in the high gamma range (>90 Hz), but this did not differ from the vehicle condition (**Supplemental Figure 5B**). We did observe a significant difference in power in the amygdala between DCZ and vehicle treatments between 20-60 Hz (within beta and gamma ranges). Power in this range increased from baseline after treatment with vehicle, but either decreased or did not differ from baseline after DCZ treatment (black dotted line, **Supplemental Figure 5B**). No differences in frequencies below 20 Hz between DCZ and vehicle conditions were observed.

Next, we looked at the relationship between LFP activity in our regions of interest by extracting the LFP coherency between amygdala and vIPFC. We first averaged coherograms across bipolar sites, sessions and periods, in order to reveal in an unbiased manner a frequency band of interest for each animal, i.e. a band in which we observed a clear peak in the coherency. For both animals, a peak in coherency was evident from 6.5 to 14.5 Hz, a range centered on the alpha band (**Supplemental Figure 5A**). Note that the previous analysis of LFP power showed that there was no change in power within the alpha band between DCZ and vehicle conditions. This is important as it indicates that any change in coherency between amygdala and vIPFC that we might detect in the alpha band cannot simply be explained by changes in power between drug treatment conditions.”

R2.M8. Can the authors rule out contributions from anesthesia? Would they expect similar results under non-anaesthetized rest conditions (which may be more clinically relevant)?

Response R2.M8: The reviewer's point is well taken. As we discuss on page 20 of the manuscript, anesthesia likely had an effect on the results, and has been shown previously to reduce some resting-state network functional connectivity. However, resting state data is highly sensitive to motion. For this reason, our group and many others attempting to collect resting state fMRI data from non-human primates have chosen to use an anesthetized paradigm to reduce the possibility of motion artifacts biasing the results (e.g. Milham et al. 2018, *Neuron*; Yacoub et al. 2020, *NeuroImage*; Autio et al. 2021, *NeuroImage*). Recent work in rodents and primates indicates that higher levels of isoflurane anesthesia cause greater reductions in resting state network activity, but that low levels of anesthesia can preserve a good deal of this activity. For this reason, we chose to use a low-dose maintenance anesthesia protocol that gave us an appropriate balance between the preservation of network activity and noise from motion. Because of how important a point this is, we have expanded the discussion of the effects of anesthesia on our data as follows:

Discussion, page 21:

“In part because of these refinements, we collected our data while animals were maintained under light ($\leq 0.9\%$) isoflurane. We chose to collect all resting state data under anesthesia due to the sensitivity of fMRI in particular to motion artifacts produced by awake animals. Although the use of anesthesia has been shown to reduce functional connectivity across resting state networks as dosage increases (Lv et al., 2016; Li and Zhang, 2017; Uhrig et al., 2018; Xu et al., 2018), the use of low-level anesthesia is a compromise that prevents motion artifacts while allowing for maximal preservation of resting state network activity (Hutchison et al., 2014; Wu et al., 2016; Areshenkoff et al., 2021; Autio et al., 2021); we consider the use of low-level isoflurane particularly important given that cortical rs-FC is most strongly impacted by increasing doses of anesthesia (Hutchison et al., 2014; Lv et al., 2016; Giacometti et al., 2022).”

Reviewer #3:

I applaud the authors for conducting such an intricate study to examine chemogenetic manipulation of the brain in a nonhuman primate model. Authors use complimentary in vivo techniques of resting-state fMRI and electrophysiology to demonstrate how activation of inhibitory DREADD receptors impact neuronal activity in fronto-limbic circuits. Specifically, the authors find that activation of the inhibitory DREADD receptor in the basolateral amygdala results in increased resting-state functional connectivity and increased local field potential coherency. These findings are paradoxical to what has been previously reported in many rodent studies suggesting that activation of inhibitory DREADD receptors decrease and inhibit neuronal activity. This study and other previous studies highlight the essential species differences between murine models and nonhuman primates and how these chemogenetic techniques are working in each model species. This also demonstrates the importance of nonhuman primate studies for translating these chemogenetic tools for use in human clinical patient populations. Overall, I thought the study was well-designed and conducted, as well as thoroughly described, but I have a few comments that I would like for authors to consider. I have

outlined my comments below:

R3.1. The authors attempt to replicate the findings from Grayson, et al, 2016 present a critical finding about the broader use of CNO in past, present, and future chemogenetic studies in rodents and nonhuman primates. Specifically, authors show that using CNO to activate the DREADD receptor results in inconsistent pattern of changes in rs-FC. Taken together the inconsistent changes in rs-FC with CNO activation and that CNO is easily metabolized into clozapine (a highly potent antipsychotic with affinity for multiple receptor types) may suggest that prior work conducted with CNO or low dose clozapine are largely due to the off-target effects of clozapine and not necessarily the DREADD receptor inhibition. In general, I feel that the authors do not emphasize this important point in their discussion of the results.

Response R3.1: We agree that this is an important point to make regarding the consistency and reliability of results obtained by using CNO as a DREADD actuator. As this comparison was not the main focus of our study, we restrict comparisons of CNO and DCZ as DREADD actuators to the discussion, but we have expanded on the current **discussion**, page 21, to more explicitly make this point:

“Why CNO activation of DREADD receptors causes such a different pattern of rs-FC remains an open question, but likely relates to the fact that this ligand is metabolized to clozapine (Gomez et al., 2017; Raper et al. 2017). Clozapine has high affinity for D1 and D2 dopamine and 5-HT_{2A} and 5-HT_{2C} serotonin receptors (Phillips et al., 1994) that are widely distributed across the brain. It is likely that the off-target effects of clozapine on these receptors underlies the wide-scale effects of DREADD inactivation of amygdala on rs-FC. **With this in mind, it may be difficult to interpret the results of studies using CNO or clozapine exclusively, as although these actuators will cause DREADD activation, distinguishing the effects of DREADDs alone from off-target activity is challenging. Further, it is likely that individual animals metabolize CNO to clozapine at differing rates, complicating interpretation. This means that some previously reported behavioral findings using DREADD receptors may actually be driven by off-target effects of CNO/clozapine. Our results lend further support to the argument that future studies using DREADDs should avoid the use of CNO or clozapine as DREADD actuators, and instead use DCZ.**”

R3.2. The discussion section lacks any discussion about what these findings mean for behavior. Authors should consider discuss their findings in light of studies in nonhuman primates that have used inhibitory DREADDs in the amygdala (Raper, et al, 2019; Roseboom, et al, 2021). Both studies found decreased behavioral responses (decreased freezing) when activating the inhibitory DREADDs, but both studies are using CNO and/or clozapine as the DREADD ligand of activation. So based on your findings in rs-FC and ephys with DCZ and CNO, are the findings from Raper and Roseboom entirely dependent on clozapine causing a decrease in activity? Would their results have been different if DCZ had been available for DREADD activity and neuronal activity would have increased instead of decreased?

- Raper J, Murphy L, Richardson R, Romm Z, Kovacs-Balint Z, Payne C, Galvan A. Chemogenetic Inhibition of the Amygdala Modulates Emotional Behavior Expression in Infant Rhesus Monkeys. *eNeuro*. 2019 Oct 14;6(5):ENEURO.0360-19.2019. doi: 10.1523/ENEURO.0360-19.2019. PMID: 31541000; PMCID: PMC6791827.

- Roseboom PH, Mueller SAL, Oler JA, Fox AS, Riedel MK, Elam VR, Olsen ME, Gomez JL, Boehm MA, DiFilippo AH, Christian BT, Michaelides M, Kalin NH. Evidence in primates supporting the use of chemogenetics for the treatment of human refractory neuropsychiatric disorders. *Mol Ther*. 2021 Dec 1;29(12):3484-3497. doi: 10.1016/j.ymthe.2021.04.021. Epub 2021 Apr 23. PMID: 33895327; PMCID: PMC8636156.

Response R3.2: As our study did not involve a behavioral measure and the animals were anesthetized during all procedures, we did not feel it was appropriate to make any larger claims about how DCZ induced inhibition of the amygdala would influence behavior. In previous studies, including Raper et al. 2019 and Roseboom et al. 2021, behavioral effects of CNO-mediated DREADD activation were compared to non-DREADD expressing control animals that also were given CNO. Although we might expect that repeating these experiments with DCZ would clarify these behavioral findings and might result in a stronger observable DREADD effect, we don't feel the findings of these studies should be entirely dismissed. We agree with the reviewer that re-assessment of the effects of DREADDs with more specific ligands will be an important next step in this research. We have added a discussion of this potential issue to the discussion section of the revised manuscript; see Response R3.1.

R3.3. Was the resting-state fMRI conducted prior to chemogenetic transfection surgery? If so, is there any difference in how simply introducing these novel receptors to the brain impact rs-FC?

Response R3.3: These animals were scanned before and after chemogenetic transfection surgery, and additionally were a part of the dataset used in our previous paper (Fujimoto et al. 2022 *J Neurosci*). In that study, we showed that DCZ injection in unoperated animals at the dose level used here (0.1 mg/kg DCZ) did not significantly alter resting state networks.

We agree however that some effect of surgery would not be completely unexpected, as the process of injecting virus into the amygdala could cause some minimal damage to the tissue. To explore this possibility, we conducted an analysis to compare the pre- and post-surgical functional connectivity maps. For both animals, we had performed two fMRI rs-FC sessions prior to surgery when VEH was injected. We therefore averaged the baseline (pre-injection) data to get an average pre-surgery rs-FC baseline. Then, we compared this pre-surgery data to the post-surgery baseline rs-FC, averaging the pre-injection data for the vehicle and DCZ sessions. We include a figure here showing an example of this comparison for one animal in one hemisphere (matching Figure 3). As can be seen in this figure there is a strong correspondence between pre- and post-

surgical scans. For instance, amygdala connectivity to frontal cortex is highly similar across the sessions and the negative functional connectivity to medial cortex as well as basal ganglia is maintained. We note that there are some differences between the pre- and post-surgery scans; rs-FC was significantly decreased in the anterior insula, amygdala, temporal lobe regions TE/TEO, and cerebellum, and was significantly increased in anterior cingulate cortex and visual cortex areas MT, V2, and V1, across both animals and hemispheres.

A note of caution is required when interpreting the above results, unlike our comparison in Figure 3, these 'baseline' data are taken from different sessions and averaged together. Therefore across session sources of variability/noise are likely to impact the results. For instance, variability likely results from differences in across-session anesthesia level, blood iron concentration, or inhomogeneities in the scanner field. Note that for the data presented in the manuscript these issues were controlled for by subtracting a within session baseline from the effect of drug administration. Because the differences are relatively small and because of the issue of across-session noise sources it is probably fair to conclude that surgery had little impact on amygdala rs-FC.

Figure R3. Representative average amygdala rs-FC across the brain before and after DREADD transfection surgery. Average rs-FC with a left amygdala seed region in animal L. **A.)** Pre-DREADD transfection surgery baseline, average of two sessions. **B.)** Post-DREADD transfection surgery baseline, average of four sessions. Scale bar indicates z-score of rs-FC. **C.)** Difference in rs-FC between post-DREADD transfection surgery baseline and Pre-DREADD transfection surgery baseline. Scale bar indicates difference in z-score of rs-FC. Cluster settings: threshold $p=0.0485$, cluster size ≥ 30 voxels, voxel faces touching.

R3.4. As a minor point, in addition to citing Gomez et al, 2017 regarding the low BBB penetrance of CNO and reverse metabolism to clozapine, authors should also cite Raper et al, 2017 because it was focused on nonhuman primates.

- Raper J, Morrison RD, Daniels JS, Howell L, Bachevalier J, Wichmann T, Galvan A. Metabolism and Distribution of Clozapine-N-oxide: Implications for Nonhuman Primate Chemogenetics. ACS Chem Neurosci. 2017 Jul 19;8(7):1570-1576. doi: 10.1021/acscchemneuro.7b00079. Epub 2017 Mar 30. PMID: 28324647; PMCID: PMC5522181.

Response R3.4: We thank the reviewer for this suggestion. We have added this reference as follows:

Page 16: "While CNO has a high affinity for the DREADD receptors, it has been found to have low blood brain barrier penetrance and is back-metabolized to clozapine, causing non-specific effects as clozapine binds to and activates endogenous receptors in the brain (Gomez et al., 2017; Raper et al. 2017)."

Page 21: "In the years since Grayson et al. was published, refinements have been made in macaque anesthetized fMRI protocols (Xu et al., 2019; Autio et al., 2021; Fujimoto et al., 2022), DREADD constructs (Galvan et al., 2019), and DREADD actuators (Gomez et al., 2017; Raper et al. 2017; Nagai et al., 2020)."

Page 21: "The latter has been shown to be metabolized to clozapine, which has broad affinity for endogenous neuromodulatory receptors in the brain in addition to DREADDs (Gomez et al., 2017; Raper et al. 2017)."

Page 21: "Why CNO activation of DREADD receptors causes such a different pattern of rs-FC is an open question, but likely relates to the fact that this ligand is metabolized to clozapine (Gomez et al., 2017; Raper et al., 2017)."

Reviewer #4:

Elouette and colleagues present a comprehensive and wonderfully executed study on the neural basis of resting state functional connectivity revealed by chemogenetics.

First of all I would like to congratulate the team on this study. It is extremely rare to see

such a comprehensive account of chemogenetic effects in NHP and the authors have gone above and beyond to describe the effects both from the perspective of fMRI and electrophysiology.

There are however some small aspects about the paper that I wish the authors could address.

R4.1: On the big picture level I find the framing of the manuscript a little bold to be honest, while the study does give us a more thorough understanding of what happens to functional connectivity in a specialized fronto-limbic circuit when it is perturbed there are no simultaneous recordings that would be needed to give a full account of the relationship between rs-FC and its neuronal substrate. Nevertheless, the framing around the fact that timescales are distinct in frontal circuits and that these distinct connectivity patterns could lead to different ???relationships??? is interesting and something the reviewer appreciates highly.

Response R4.1: The reviewer's point about the lack of simultaneous recordings is well taken. Consequently, we have added the following clarification to the manuscript to highlight that our findings do not come from simultaneous recording and thus the link between levels of analysis requires further investigations, and these findings may not generalize to all amygdala-frontal connections:

Discussion, page 18:

“One important caveat about the current design is that the macroscale functional neuroimaging and micro/meso-scale electrophysiological recordings were collected in separate sessions. Simultaneous fMRI and electrophysiological recordings could allow a direct comparison between the modalities, but there are many technical barriers associated with crosstalk between the two, particularly when considering single neuron spiking data (Vanduffel et al., 2014). However, the use of the same paradigm performed in the same subjects across both conditions has been previously used to provide a link between fMRI and the underlying neural activity (Park et al., 2017, 2022). Our experimental design makes use of a within subject design across the fMRI and extracellular recording experiments, allowing for comparison of results across both paradigms while minimizing noise from variability across subjects. Consequently, while it is not possible to directly link specific features of fMRI and neural activity signals to each other (i.e. spiking in the amygdala to fMRI fluctuations), our design allows for correlational links between local and systems level effects. Additionally, our work provides a foundation from which future studies may more directly examine the relationship between fMRI rs-FC and neural activity in limbic and frontal areas.”

R4.2: One thing worth discussing in the paper is the fact that a non specific DREADDs was used and this likely affects the results. I understand why this approach was chosen but the implications would be worth discussing by the authors.

Response R4.2: We chose this construct after weighing the pros and cons of a more specific approach, namely that this construct was one of the most well-validated available and had a better chance of achieving strong transfection in our area of interest, as opposed to a dual-transfection strategy where slight variations in both injection sites can have a large effect on the transfection. However, there are drawbacks to our approach as the reviewer notes, as our virus was specific only for neurons and did not distinguish between, for example, inhibitory or excitatory neurons in amygdala (see response to R2.4). We agree that this an important point to highlight as it is important to take into account when interpreting the effects, and have added the following to the **discussion**, page 20:

“Only three recent studies in macaques have recorded spiking activity from neurons after DREADD transfection in the target region (Deffains et al., 2021; Hasegawa et al., 2022; Perez et al., 2022), and all observed a decrease in firing rate after chemogenetic inhibition of the target. **However, these studies made use of constructs that targeted various populations of cells (respectively, all neurons, all cells, and all catecholaminergic neurons), none recorded from amygdala, and none of these targeted fronto-limbic circuitry. Our own approach targeted all neurons, and so our transfected cells included both inhibitory and excitatory neurons in the amygdala (Supplemental Figure 1). Although this makes our inhibitory chemogenetic approach more comparable to the effects of pharmacological inhibition or excitotoxic lesion, it does alter the interpretation of the effects on fMRI and neurophysiology. To this point, our finding that activating an inhibitory DREADD in macaque amygdala led to increased spiking in this area (Figure 5) could be related to the composition of this structure in primates. Recent studies have shown that there is an increased diversity of interneurons in macaque amygdala compared to rodents (Beyeler and Dabrowska, 2020; McDonald and Augustine, 2020; McDonald, 2021). Thus, we speculate that inhibiting these cells may have caused a paradoxical disinhibition of local amygdala circuits, which led to increased extracellular activity. A second possibility is that the inhibition of local excitatory neurons that drive inhibitory networks within the amygdala could have led to increased local activity (Smith et al., 2000). These explanations are not mutually exclusive, and future work may serve to clarify the mechanisms at play. The use of DREADD constructs specific for excitatory neurons, or DREADD constructs that can target specific projections between two areas, would be a valuable next step.”**

R4.3: Overall this is a highly significant paper which is very well written and thoroughly executed. The analysis is rigorous and the results stand on their own. The paper should be accepted with very minor revisions.

Response R4.3: We thank the reviewer for their support for our work.

REVIEWERS' COMMENTS

Reviewer #2 (Remarks to the Author):

Thank you for addressing all my previous questions. Congratulations to this fantastic work-

Reviewer #3 (Remarks to the Author):

Again I applaud the authors for conducting such an intricate study to examine chemogenetic manipulation of the brain in a nonhuman primate model. I appreciate the authors thoroughly addressing all of my concerns. Their edits to my and other reviewer comments have greatly improved the manuscript. I have no further comments at this time. Overall, I believe their data demonstrate the importance of nonhuman primate studies for translating these chemogenetic tools for use in human clinical patient populations.

Reviewer #4 (Remarks to the Author):

The authors have thoroughly responded to my comments and expanded the manuscript significantly making it even stronger.